

**A novel method for correcting water budget components and reducing their**
**uncertainties by optimally distributing the imbalance residual without full**
**closure**
Zengliang Luo[a,b], Hanjia Fu[a,b], Quanxi Shao[c]*, Wenwen Dong[a,b], Xi Chen[d], Xiangyi
Ding[e]*, Lunche Wang[a,b], Xihui Gu[a,b], Ranjan Sarukkalige[f], Heqing Huang[a,b], Huan Li[g]
[a] Hubei Key Laboratory of Regional Ecology and Environmental Change, State Key
Laboratory of Geomicrobiology and Environmental Changes, China University of
Geosciences, Wuhan 430074, China.
[b] State Key Laboratory of Biogeology and Environmental Geology, China University
of Geosciences, Wuhan 430074, China.
[c] CSIRO Data61, Australian Resources Research Centre, 26 Dick Perry Avenue,
Kensington, WA 6155, Australia.
[d] State Key Laboratory of Remote Sensing Science, Aerospace Information Research
Institute, Chinese Academy of Sciences, Beijing, 100101, China.
[e] Department of Water Resources, China Institute of Water Resources and
Hydropower Research, Beijing 100038, China.
[f] School of Civil and Mechanical Engineering, Curtin University, GPO Box U1987,
Perth, WA 6845, Australia.
[g] HUN-REN Balaton Limnological Research Institute, Hungary.
Corresponding author to: Quanxi Shao (Quanxi.Shao@data61.csiro.au) and Xiangyi
Ding (dingxy@iwhr.com)
**Highlights:**
• There are around 5% (with range between 0 to 10%) of cases where errors
introduced by current BCC methods are so large that some budget components
become negative
• A Novel IWE-Res method is proposed to identify the optimal balance for
redistributing ΔRes
• The optimal redistribution of ΔRes is found between 40%-90% in most basins,
except for cold regions.
**Abstracts:** Closing the water budget improves the consistency of water budget
component datasets, including precipitation (P), evapotranspiration (ET), streamflow





(Q) and terrestrial water storage change (TWSC), thereby enhancing the
understanding of basin-scale water cycle processes. Existing water budget closure
correction (BCC) methods typically redistribute the entire water imbalance error
($\Delta$Res) to achieve perfect water budget closure but often neglect the trade-off between
achieving closure and the errors introduced into budget components as a result of this
redistribution. This study quantifies the uncertainties introduced by existing BCC
methods (CKF, MCL, MSD, and PR) across 84 basins representing diverse climate
zones. We then propose a novel method, IWE-Res, to identify the optimal balance for
redistributing $\Delta$Res. This method minimizes the combined error from both introduced
budget component errors and the remaining $\Delta$Res error, while reducing the occurrence
of negative values. The results indicate: (1) Existing BCC methods can lead to
negative values in corrected budget components, with negative values comprising
approximately 0–10% (mostly below 5%) of the time series; (2) Compared to existing
BCC methods, the proposed IWE-Res method improves the accuracy of corrected P
by 29.5%, corrected ET by 24.7%, corrected Q by 69.0%, and corrected TWSC by 6.8%
based on the root mean square error (RMSE); and (3) In most basins, except in cold
regions, the optimal balance is reached when 40%–90% of $\Delta$Res is redistributed. By
offering a more balanced approach to water budget closure, this study improves the
accuracy and reliability of corrected budget component datasets.
**Keywords:** Water budget closure; Budget components; Water imbalance; Uncertainty
identification; Global hydrology



**1 Introduction**

Closing the water budget is essential for understanding water circulation among the atmosphere, surface, soil, and groundwater (Li et al., 2024; Mourad et al., 2024). However, the absence of integrated observational systems capable of measuring all water budget components simultaneously—since these components are typically observed separately—makes it challenging to obtain datasets that achieve water budget closure (Eq. 1) through direct observations (Zheng et al., 2025).

$$P - ET - Q - TWSC = 0 \tag{1}$$

where P represents precipitation, ET represents evapotranspiration, Q represents streamflow, and TWSC represents terrestrial water storage change.

In research and applications, hydrological models are designed based on the principle of water balance. However, extensive simplifications and error propagation within these models (arising from input data, model structure, and parameter uncertainties) introduce errors in the simulation of budget components, making water budget closure equally challenging. In the era of big data, the growing availability of remote sensing and reanalysis datasets offers greater potential for achieving water budget closure (Zhou et al., 2024). To correct water imbalance error (ΔRes) and ensure ΔRes = 0 (where ΔRes = P – ET – Q – TWSC), various water budget closure correction (BCC) methods have been proposed and widely adopted. Common methods include Proportional Redistribution (PR), the Constrained Kalman Filter (CKF), Multiple Collocation (MCL), and the Minimized Series Deviation (MSD) method (Pan et al., 2012; Luo et al., 2023). For example, Abhishek et al. (2021)



applied the PR, CKF, and MCL methods to quantify water budget closure and
uncertainties in budget components in the upper Chao Phraya River basin;
Abolafia-Rosenzweig et al. (2021) evaluated the effectiveness of PR, CKF, and MCL
methods in closing the water budget for 24 global basins; Dastjerdi et al. (2024)
developed a precipitation data merging method to improve precipitation estimates
based on existing BCC methods.

Existing BCC methods achieve water budget closure by redistributing the entire

ΔRes error across budget components, with redistribution weights estimated based on
errors in these components. However, ΔRes represents a composite error that includes
not only inaccuracies in measured components but also contributions from
unmeasured components. The latter is prevalent but difficult to attribute to specific
budget components using existing technologies. Consequently, existing BCC methods
determine redistribution weights solely based on estimated errors in budget
components. Since existing BCC methods address only ΔRes errors arising from
inaccuracies in measured components, they inherently conflict with the goal of
achieving a fully closed water budget. This limitation explains why, despite aiming to
improve the accuracy of P, ET, Q, and TWSC estimates through complete
redistribution of ΔRes, existing BCC methods may lead to limited improvements—or
even a decline—in the accuracy of corrected budget component datasets.

A clear manifestation of this issue is the occurrence of negative values in

corrected budget component datasets, such as negative P, ET, and Q. Our previous
work also found that enforcing water budget closure may, to some extent, reduce the





accuracy of budget components and tends to introduce an ET regulation factor to
mitigate accuracy loss in ET caused by existing BCC methods (Luo et al., 2023). A
more effective approach to addressing this issue may involve identifying an optimal
distribution of the ΔRes error that balances errors introduced in budget components
with the remaining ΔRes error. Specifically, ΔRes errors arising from inaccuracies in
budget components should be redistributed while preventing the negative values
associated with existing BCC methods.
The key question we aim to answer in this study is the extent of uncertainty
introduced into budget components by existing BCC methods for enforcing water
budget closure and, more critically, whether this uncertainty exceeds the reduction in
the ΔRes error. If the introduced uncertainty outweighs the error reduction, fully
closing the water budget may be unnecessary. As noted earlier, ΔRes represents a
composite error, whereas existing BCC methods primarily address errors in budget
components. We propose that an optimal balance for redistributing the ΔRes error
should be identified—one that minimizes the combined error from budget
components and the remaining water imbalance. This optimal balance allows for
redistributing only the portion of ΔRes attributable to errors in budget components,
rather than the entire ΔRes, thereby preventing the occurrence of negative values in
budget components due to improper error redistribution. However, research on
identifying this optimal balance, which is crucial for improving existing BCC
methods, remains lacking.
The primary objective of this study is to quantify the uncertainties introduced by





existing BCC methods in closing the water budget from multiple perspectives and to
propose a new IWE-Res method for identifying the optimal balance in ΔRes
redistribution. To enhance the robustness of error analysis and validate the proposed
IWE-Res method, we applied four existing BCC methods with varying principles and
complexities (PR, CKF, MCL, and MSD) across 84 global basins with diverse
climatic characteristics. The specific objectives of this study are:
(1) To quantify the uncertainties introduced into budget components by enforcing
water budget closure using existing BCC methods from multiple perspectives,
including uncertainties relative to observations, the occurrence of negative values in
budget components, and deviations from the original budget component datasets. This
analysis provides a more comprehensive understanding of the trade-offs between
achieving water budget closure and the associated errors;
(2) To analyze in detail the occurrence of negative corrected values in budget
components caused by existing BCC methods, including the proportion of negative
values within the time series of each budget component and their spatial distribution
under varying climatic conditions;
(3) To compare the reduction in ΔRes with the corresponding increase in budget
component errors resulting from enforced water budget closure;
(4) To propose a new method (IWE-Res) for identifying the optimal balance in
ΔRes redistribution, minimizing the combined error from both introduced budget
component errors and the remaining ΔRes error. The accuracy and reliability of the
proposed IWE-Res method were validated through comparisons with existing BCC



144 methods (PR, CKF, MCL, MSD).

**2 Study area and data**

146 To robustly quantify the uncertainties introduced by existing BCC methods in

147 closing the water budget and to assess the accuracy of the proposed IWE-Res method

148 across different climate zones, multiple river basins worldwide were selected as study

149 areas. In total, 84 basins (Fig. 1) were chosen based on the availability of streamflow

150 observations from the GRDC for the period 2002–2020. To ensure data reliability, the

151 proportion of missing data was kept below 10%, with missing values interpolated

152 using a linear method. Notably, approximately 90% of the basins used in this study

153 had less than 5% missing data.

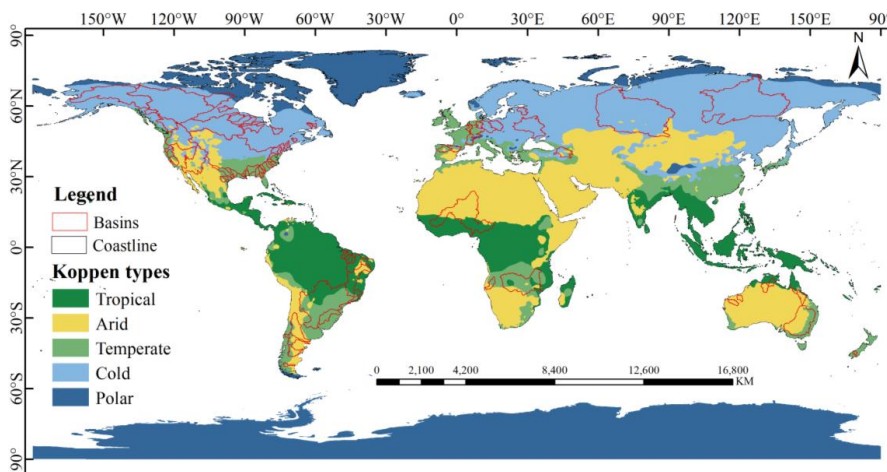


155 **Fig. 1** Overview of the Study Area. The climate classification used in this study is

156 based on the Köppen climate classification system.


158 The climate classifications presented in Fig. 1 were determined using the Köppen

159 climate classification system, a widely adopted framework that categorizes global



climates based on temperature and precipitation thresholds (Crosbie et al., 2012;
Hansford et al., 2020; Liu et al., 2022; Papacharalampous et al., 2023). This system
divides the world into five primary climate types—Tropical, Arid, Temperate, Cold,
and Polar. Its key strength lies in its integration of climate data with vegetation
distribution, making it highly relevant to ecological environments.

For each budget component, multiple datasets are typically available, with

accuracy varying across different basins. No single dataset consistently performs best
across all global basins. Therefore, multiple datasets were selected for each budget
component to generate various data combinations (Equations 2–3). This approach
ensures the inclusion of the most suitable dataset combinations while mitigating
uncertainties associated with reliance on a single dataset.

In selecting datasets, priority was given to those incorporating extensive

observational data, as they generally offer higher accuracy. We selected four P
datasets—GPCC, GPM, MSWEP, and PERSIANN-CDR; three ET datasets—GLDAS,
GLEAM, and TerraClimate; and three TWSC datasets derived from GRACE satellite
observations—GRACE CSR, GRACE GFZ, and GRACE JPL. Observed Q data were
obtained from the GRDC platform. By combining these datasets, a total of 36 distinct
data combinations were generated for each basin (Equation 3).

$Cjkl = \begin{bmatrix} P_j & ET_k & TWSC_l & Q \end{bmatrix}$                      (2)

where $j$, $k$, and $l$ represent the indices of the datasets corresponding to each budget
component. Table 1 provides basic information on the datasets used in this study,
along with their corresponding indices. Equation 3 represents a matrix composed of



the elements defined in Equation 2.
$$C = \begin{bmatrix} C111 & C112 & C113 & C121 & C122 & C123 & C131 & C132 & C133 \\ C211 & C212 & C213 & C221 & C222 & C223 & C231 & C232 & C233 \\ C311 & C312 & C313 & C321 & C322 & C323 & C331 & C332 & C333 \\ C411 & C412 & C413 & C421 & C422 & C423 & C431 & C432 & C433 \end{bmatrix} \quad (3)$$

The Global Precipitation Climatology Centre (GPCC) dataset, operated and
provided by the German Weather Service (DWD), is a global precipitation dataset
based on ground station observations (Becker et al., 2013; Schneider et al., 2008).
This dataset is generated through the quality control, spatial interpolation, and
aggregation of observational data from various sources, resulting in a consistent,
globally comprehensive precipitation product. It features a spatial resolution of 0.25°
and is available at both daily and monthly time scales. The Global Precipitation
Measurement Integrated Multi-Satellite Retrievals (GPM IMERG) dataset, initiated
by NASA and JAXA, provides global precipitation estimates by integrating satellite
sensor data with ground-based rain gauge observations and other auxiliary data,
ensuring thorough calibration. The Multi-Source Weighted-Ensemble Precipitation
(MSWEP) dataset is produced by optimally combining precipitation data from
satellite observations, ground stations, and reanalysis products. It applies different
weighting strategies across varying spatial and temporal scales to maximize data
accuracy (Beck, Pan, et al., 2019; Beck et al., 2017; Beck, Wood, et al., 2019). The
PERSIANN-CDR dataset, derived from satellite remote sensing and artificial neural
network technology, covers latitudes from 60°S to 60°N, with a spatial resolution of
0.25° and daily temporal resolution.
The Global Land Data Assimilation System (GLDAS), jointly developed by



NASA and NOAA, employs advanced land surface modeling and data assimilation
techniques to integrate satellite and ground-based observations, generating optimal
fields of land surface states and fluxes. The GLDAS dataset has provided continuous
global evapotranspiration data since 2000. The GLEAM evapotranspiration dataset,
developed by the Miralles team at the University of Bristol, estimates global surface
evapotranspiration using multi-satellite remote sensing data and the Priestley-Taylor
retrieval algorithm. Potential evaporation is calculated based on observations of
surface   net   radiation   and   near-surface   air   temperature.   The   potential
evapotranspiration estimates are converted to actual evapotranspiration using a
multiplicative evaporative stress factor, derived from microwave-based Vegetation
Optical Depth (VOD) and root zone soil moisture estimates. TerraClimate utilizes the
Penman-Monteith   equation   to   estimate   both   global   potential   and   actual
evapotranspiration. To improve accuracy, TerraClimate's evapotranspiration estimates
are calibrated against ground-based observations and other climate data products,
ensuring applicability across various climate regions and seasons (Abatzoglou et al.,

2018).

The launch of gravity satellites has provided new opportunities for more accurate

observations of large-scale TWSC in river basins. The Gravity Recovery and Climate
Experiment (GRACE) was launched in March 2002 and completed its mission in
November 2017. Its successor, GRACE Follow-On (GRACE-FO), was launched in
May 2018, continuing the monitoring of Earth's gravity field and its changes
(Boergens et al., 2024). The principle behind observing TWSC is the assumption that



variations in terrestrial gravity are primarily caused by water mass changes. By
tracking fluctuations in Earth's gravity field, information on the distribution and
changes in surface water can be inferred. The GRACE TWSC datasets used in this
study are provided by the University of Texas Center for Space Research (CSR), the
German Research Centre for Geosciences (GFZ), and NASA's Jet Propulsion
Laboratory (JPL).

The Global Runoff Data Centre (GRDC) provides the most comprehensive

open-access river discharge data available worldwide, collected from national
hydrological agencies. This dataset includes river streamflow measurements from
over 10,000 stations across 159 countries (Su et al., 2024). To minimize the impact of
missing data on the reliability of the results, hydrological stations were selected based
on the criterion that missing values accounted for less than 10% of the total dataset.
Linear interpolation was then applied to fill any remaining data gaps.
**Table 1** Datasets used for this study.

| Variable | Data Source | Number | Resolution | Reference |
|---|---|---|---|---|
| P | Global Precipitation Climatology Centre (GPCC) | 1 | 0.25°/month | Schneider et al. (2008) |
| | Global Precipitation Measurement (GPM) | 2 | 0.1°/month | Huffman et al. (2015) |
| | Multi-Source Weighted-Ensemble Precipitation (MSWEP) | 3 | 0.1°/month | Beck, Wood, et al. (2019) |
| | Precipitation Estimation from Remotely Sensed Information using Artificial Neural Networks — Climate Data Record (PERSIANN-CDR) | 4 | 0.25°/month | Hsu et al. (1997) |
| ET | global land data assimilation system (GLDAS) | 1 | 0.25°/month | Park and Choi (2015) |
| | Global Land Evaporation Amsterdam Model (GLEAM) | 2 | 0.25°/month | Miralles et al. (2011) |



| | TerraClimate | 3 | 1/24°/month | Abatzoglou et al. (2018) |
|---|---|---|---|---|
| TWSC | Gravity Recovery and Climate Experiment (GRACE CSR) | 1 | 1.0°/month | Watkins et al. (2015) |
| | Gravity Recovery and Climate Experiment (GRACE GFZ) | 2 | 1.0°/month | Watkins et al. (2015) |
| | Gravity Recovery and Climate Experiment (GRACE JPL) | 3 | 1.0°/month | Watkins et al. (2015) |
| Q | Global Runoff Data Centre (GRDC) | - | - | Burek and Smilovic (2022) |


## 3 Methods

*3.1 Water imbalance error*

The water balance equation describes the conservation of mass between water inflows, outflows, and changes in storage within a given region (Equation 1). However, due to measurement errors, model simplifications, and unmeasured components (omissions), the observed budget components contain errors (named as $\varepsilon_P$, $\varepsilon_{ET}$, $\varepsilon_Q$, $\varepsilon_{TWSC}$, respectively), causing the water balance to be unclosed (Equation 1 becomes Equation 4) (Aires, 2014; Wong et al., 2021). The imbalance resulting from these errors is defined as the $\Delta Res$ error (Equation 5), representing inconsistencies among budget components.

Minimizing the $\Delta Res$ error is a key objective in practical hydrological applications, as it enhances the accuracy and reliability of budget component datasets. However, it is important to note that smaller $\Delta Res$ values may arise from error compensation among budget components rather than genuine improvements in data accuracy. Therefore, a high-precision water balance dataset is characterized not only by a near-zero $\Delta Res$ error but also by budget components that closely approximate



their true values (Luo et al., 2021).
$(P + \varepsilon_P) - (ET + \varepsilon_{ET}) - (Q + \varepsilon_Q) - (TWSC + \varepsilon_{TWSC}) = 0$    (4)
$\Delta Res = \varepsilon_{ET} + \varepsilon_Q + \varepsilon_{TWSC} - \varepsilon_P = P - ET - Q - TWSC$   (5)
where $\varepsilon_P$, $\varepsilon_{ET}$, $\varepsilon_Q$, $\varepsilon_{TWSC}$ are the errors in budget components of P, ET, Q, and
TWSC relative to their true values, respectively.
*3.2 Existing water budget closure correction methods*
To minimize the $\Delta Res$ error in Equation 5 (reducing $\Delta Res$ from $\neq 0$ to 0), various
statistical BCC methods have been developed. These methods differ in their principles
to redistributing the $\Delta Res$ error, leading to varying levels of introduced uncertainty. To
systematically assess the uncertainties associated with existing BCC methods in
closing the water budget and to reduce uncertainty in method selection, we evaluated
four representative methods: PR, CKF, MCL, and MSD (Luo et al., 2023;
Abolafia-Rosenzweig et al., 2021; Dastjerdi et al., 2024).
For each basin, these four methods were applied to 36 different data
combinations (Equation 3), yielding 144 uncertainty estimates. The optimal
combinations were identified using a 5% threshold. By averaging the errors
introduced into budget components across these selected optimal combinations, we
quantified the uncertainty associated with existing BCC methods. This approach
minimizes uncertainties arising from both BCC method selection and budget
component data selection, enabling a more objective evaluation of the errors
introduced by existing BCC methods. A brief overview of the four BCC methods is
provided below:





(1) PR method

The PR method assumes that the error in budget components is proportional to

their magnitudes (Abatzoglou et al., 2018). Based on the relative magnitudes of these
variables, the ΔRes error is redistributed across them to achieve water budget closure
(Equation 6).
$$F_i = X_i - \Delta Res(G_i)\left(\frac{|X_i|}{\sum_{j=1}^{n}|X_j|}\right)$$
(6)

where $F_i$ and $X_i$ represent the corrected and original data for budget components (P,
ET, Q and TWSC), respectively; n denotes the number of budget components
involved in the water budget closure calculation; $\Delta Res$ represents the water
imbalance error; $G$ is a constant vector defined as $G = [1\ -1\ -1\ -1]$.

(2) CKF method

The CKF method is developed based on the Kalman filter. For a given set of

estimated budget components $X = [P\ ET\ Q\ TWSC]^T$ and their estimated errors
$\Delta Res = GX \neq 0$ (where G is a constant vector, $G = [1\ -1\ -1\ -1]$), the goal is
to find a new set of estimates $F = [P'\ ET'\ Q'\ TWSC']^T$ such that $GX' = 0$,
achieving water budget closure (Pan et al., 2012). In simple terms, the CKF method
redistributes the $\Delta Res$ among the budget components based on the error covariance
of $X$, defined as $\Delta\varepsilon_{XX}$ (Equation 7), to obtain a closured dataset (Equation 9).
$$\Delta\varepsilon_{XX} = \overline{(X - X_0)(X - X_0)^T}$$
(7)

$$\Delta\varepsilon_{XX} = \begin{bmatrix} \Delta\varepsilon_{P-P} & \Delta\varepsilon_{P-ET} & \Delta\varepsilon_{P-Q} & \Delta\varepsilon_{P-TWSC} \\ \Delta\varepsilon_{ET-P} & \Delta\varepsilon_{ET-ET} & \Delta\varepsilon_{ET-Q} & \Delta\varepsilon_{ET-TWSC} \\ \Delta\varepsilon_{Q-P} & \Delta\varepsilon_{Q-ET} & \Delta\varepsilon_{Q-Q} & \Delta\varepsilon_{Q-TWSC} \\ \Delta\varepsilon_{TWSC-P} & \Delta\varepsilon_{TWSC-ET} & \Delta\varepsilon_{TWSC-Q} & \Delta\varepsilon_{TWSC-TWSC} \end{bmatrix}$$
(8)

where $X_0$ refers to the reference values of the estimated budget component. The



dimension of $\Delta\varepsilon_{XX}$ is 4×4, representing the covariance of errors among the budget
components (Equation 8).
$$F = X + K(0 - GX) \tag{9}$$
where $K = \Delta\varepsilon_{XX}C^{T}(C\Delta\varepsilon_{XX}C^{T})^{-1}$ is the Kalman gain. Setting $C\hat{X} = \widehat{\Delta Res}$, and
Equation 9 can be rewritten as Equation 10.
$$F = X - \Delta\varepsilon_{XX}G^{T}(G\Delta\varepsilon_{XX}G^{T})^{-1}\Delta Res \tag{10}$$
(3) MCL method
The MCL method is an extension of the triple collocation (TC) method. It
calculates the weights for redistributing the $\Delta$Res error among budget components by
estimating the errors relative to their true values (expressed as distances, without
requiring knowledge of the true values). The fundamental equations of the MCL
method are shown in Equations 11-12.
$$F_i = X_i - \Delta Res(G^i)(d_{xx_0-norm}^i) \tag{11}$$
$$d_{xx_0-norm}^i = \frac{d_{xx_0}^i}{\sum_{j=1}^{4}\left|d_{xx_0}^j\right|} \tag{12}$$
In these equations, $F_i$ represents the corrected data for the $i$-th budget
component; $X_i$ denotes the original data for the $i$-th budget component; $\Delta$Res
represents the water imbalance error; $d_{xx_0-norm}^i$ represents the weight assigned to
the $i$-th budget component, and $d_{xx_0}^i$ represents the distance between the $i$-th
budget component and the true value, as calculated using the Monte Carlo (MC)
method. For example, in the case of five precipitation data products ($N = 5$), the
calculation of $d_{xx_0}^i$ ($d1t$, $d2t$, $d3t$, $d4t$, and $d5t$) is shown in Equations 13-14.
$$A_{(N)}y_{(N)} = b_{(N)} \tag{13}$$





$$A_{(5)} = \begin{bmatrix} 1 & 1 & 0 & 0 & 0 \\ 1 & 0 & 1 & 0 & 0 \\ 1 & 0 & 0 & 1 & 0 \\ 1 & 0 & 0 & 0 & 1 \\ 0 & 1 & 1 & 0 & 0 \\ 0 & 1 & 0 & 1 & 0 \\ 0 & 1 & 0 & 0 & 1 \\ 0 & 0 & 1 & 1 & 0 \\ 0 & 0 & 1 & 0 & 1 \\ 0 & 0 & 0 & 1 & 1 \end{bmatrix}, \ y_{(5)} = \begin{bmatrix} d_{1t}^2 \\ d_{2t}^2 \\ d_{3t}^2 \\ d_{4t}^2 \\ d_{5t}^2 \end{bmatrix}, \ b_{(5)} = \begin{bmatrix} d_{12}^2 \\ d_{13}^2 \\ d_{14}^2 \\ d_{15}^2 \\ d_{23}^2 \\ d_{24}^2 \\ d_{25}^2 \\ d_{34}^2 \\ d_{35}^2 \\ d_{45}^2 \end{bmatrix}$$  (14)

(4) MSD method
The MSD method redistributes the ΔRes to each budget component based on
minimizing the time-series deviation error, aiming to reduce model uncertainties
caused by errors in estimating time-point deviations (Luo et al., 2023). Specifically,
the MSD method first calculates the minimum time-series deviation distance between
remote sensing data for budget components and multi-source integrated data products
(EO) (Equation 15).
$$D_{x,\rightarrow n} = -\frac{\left[\sum_{j=1}^{n}(y_{(EO,j)} - \overline{y_{(EO,\rightarrow n)}})(x_{(RS,j)} - \overline{x_{(RS,\rightarrow n)}})\right]^2}{\sum_{j=1}^{n}(x_{(RS,j)} - \overline{x_{(RS,\rightarrow n)}})^2} + \sum_{j=1}^{n}(y_{(EO,j)} - \overline{y_{(EO,\rightarrow n)}})^2$$  (15)
where $D_{x,\rightarrow n}$ represents the minimum time-series deviation distance for budget
component $x$ (e.g., P, ET, TWSC); $y_{(EO,j)}$ and $x_{(RS,j)}$ refer to the integrated value
and raw value of the budget component $x$, respectively; $\overline{y_{(EO,\rightarrow n)}}$ and $\overline{x_{(RS,\rightarrow n)}}$
denote the average deviation of budget component $x$ from the first to the $n$-th time
point.
Next, the MSD method calculates the weights for each budget component based
on $D_{x,\rightarrow n}$ (Equation 16).





$$w_{x,j} = \frac{D_{x,\rightarrow j}}{\sum_{i=1}^{4} D_{i,\rightarrow j}} \qquad (16)$$

where $w_{x,j}$ is the weight of budget component $x$ at time point $j$.

Finally, the weight calculation results from Equation 16 are substituted into

Equation 17 to achieve water budget closure.

$$\begin{bmatrix} F_{P,j}^{BCC} \\ F_{ET,j}^{BCC} \\ F_{R,j}^{BCC} \\ F_{TWSC,j}^{BCC} \end{bmatrix} = \begin{bmatrix} F_{P,j}^{Raw} \\ F_{ET,j}^{Raw} \\ F_{R,j}^{Raw} \\ F_{TWSC,j}^{Raw} \end{bmatrix} - \Delta Res \begin{bmatrix} 1 \\ -1 \\ -1 \\ -1 \end{bmatrix} \begin{bmatrix} w_{P,j} \\ w_{ET,j} \\ w_{R,j} \\ w_{TWSC,j} \end{bmatrix} \qquad (17)$$

where $F^{BCC}$ represents the budget components (P, ET, Q, and TWSC) corrected for

water budget closure, while $F^{Raw}$ denotes the raw, uncorrected values of the budget

components.

*3.3 Uncertainties introduced by existing BCC methods for closing water budget*

When the existing BCC methods described in Section 3.2 are applied to close the

water budget, they redistribute ΔRes based on the estimated errors of budget

components but neglect unmeasured components. This inevitably leads to an

unreasonable redistribution of the ΔRes error, introducing new uncertainties. The

magnitude of these introduced errors and whether they can be ignored remain

unresolved, primarily due to insufficient observational data for some budget

components, making it difficult to quantify the associated uncertainties.

Our analysis in this study reveals that when existing BCC methods are used for

water budget closure, certain budget components that typically have positive values,

such as P, ET, and Q, occasionally become negative in some months. Previous studies

have also mentioned this issue (Lehmann et al., 2022). This clearly indicates an

unreasonable redistribution of ΔRes errors, underscoring the urgent need for



methodological improvements. Despite this issue, research on negative values remains
limited. Key questions persist regarding the proportion of negative values in each
budget component under current BCC methods, which variables are most susceptible
to severe negative values, and how these errors vary throughout the year. Addressing
these questions is critical for refining existing BCC methods.

Notably, quantifying negative values does not require observational data. To

comprehensively assess the uncertainties introduced by forced water budget closure,
we consider three aspects: errors of individual budget components relative to observed
values (Section 4.2.1), negative values arising from budget closure (Section 4.2.2),
and ensemble errors (Section 4.2.3).

(1) Errors of individual budget components

Quantifying this type of error requires determining reference values for budget

components. However, for certain variables, such as ET, observational data are
insufficient across global watersheds, posing a major challenge in accurately
characterizing global ET patterns. As a result, approximate reference values must be
used to ensure the reliability of the results.

In this study, reference values for budget components were established based on

the following principles. For Q, long-term observational records from hydrological
stations were available for all selected basins, meeting the study's requirements. For
TWSC, we utilized three observational datasets from the GRACE satellite, which
currently provides the only large-scale measurements of basin water storage changes
under rigorous quality control. The reliability of GRACE data has been validated





through ground-based observations (Famiglietti et al., 2011; Landerer et al., 2020;
Rodell et al., 2009; Tapley et al., 2004; Yeh et al., 2006). Thus, GRACE TWSC data
can be considered approximately reliable. To further enhance its accuracy, we applied
data fusion techniques, as described in Equation 18, to merge the three GRACE
TWSC products into a single reference dataset (Munier & Aires, 2018; Zhang et al.,

2018).

The uncertainty introduced by existing BCC methods for precipitation was

evaluated from two perspectives. First, 13 basins with sufficient observational
precipitation data were selected, using observed precipitation as the reference. This
sample size was sufficient for assessing the uncertainties associated with existing
BCC methods. Second, 71 additional basins lacking sufficient observational
precipitation data were included, for which fused precipitation values, derived using
Equation 18, served as reference. This approach enabled cross-validation of the
reliability of the fused dataset by comparing results with those from basins with
observational data, allowing the study to be extended to a larger number of basins.

ET is the most challenging budget component to measure directly. The scarcity

of globally available ET observational data precludes the direct use of observed ET as
a reference. To address this limitation, previous studies have either focused on a few
basins with available observational data or compared multiple existing ET datasets.
ET products are generally considered reliable if their magnitudes and trends align
with those of other datasets (Chen et al., 2021; Pan et al., 2020; Xu et al., 2019). Some
studies have also employed the fusion of multiple data products as a reference for ET



validation (Jiménez et al., 2018; Mueller et al., 2011; Yao et al., 2014). Following this
approach, we assessed the uncertainty introduced by existing BCC methods for ET by
utilizing a fusion-based reference dataset.
$$\overline{M_x} = \sum_{i=1}^{n} M_{x,i} * \omega_i \ and \ \omega_i = \frac{1}{\sigma_i^2} / \sum_{i=1}^{1} \frac{1}{\sigma_i^2}$$   (18)
where $\overline{M_x}$ represents the fused value of the budget component, $M_{x,i}$ denotes the
$i$-th product of the budget component; $\omega_i$ denotes the weight of the $i$-th product, and
$\sigma_i^2$ refers to the covariance error of the $i$-th product, $n$ is the total number of budget
components, and $x$ refers to P, ET or TWSC.
After establishing reference values for budget components, we quantify errors in
the original data relative to these references, using the positive metric CC and inverse
metric RMSE as examples, denoted as $CC_1$ and $RMSE_1$, respectively. Similarly, errors
in the BCC-corrected data relative to the reference values are calculated, represented
as $CC_2$ and $RMSE_2$.
To assess the uncertainties introduced by water budget closure, changes in CC
and RMSE (CC′ and RMSE′) are computed using Equations 19 and 22. Positive
values of CC′ and RMSE′ indicate an improvement in data accuracy following BCC
correction, whereas negative values suggest a decline. In addition to CC and RMSE,
other statistical metrics used in this study include the positive indicator NSE and the
negative indicator MAE.
$$CC' = CC_2 - CC_1$$   (19)
$$NSE' = NSE_2 - NSE_1$$   (20)
$$MAE' = MAE_1 - MAE_2$$   (21)




$$RMSE' = RMSE_1 - RMSE_2 \tag{22}$$

$$CC = \frac{\sum_{i=1}^{n}(Obs_i - \overline{Obs})(Sim_i - \overline{Sim})}{\sqrt{\sum_{i=1}^{n}(Obs_i - \overline{Obs})^2}\sqrt{\sum_{i=1}^{n}(Sim_i - \overline{Sim})^2}} \tag{23}$$

$$NSE = 1 - \frac{\sum_{i=1}^{n}(Sim_i - Obs_i)^2}{\sum_{i=1}^{n}(Obs_i - \overline{Obs})^2} \tag{24}$$

$$MAE = \frac{1}{n}\sum_{i=1}^{n}|Sim_i - Obs_i| \tag{25}$$

$$RMSE = \sqrt{\frac{1}{n}\sum_{i=1}^{n}(Obs_i - Sim_i)^2} \tag{26}$$

where $Obs_i$ represents the reference value at time $i$, and $Sim_i$ represents either the original data or the BCC-corrected data. $\overline{Obs}$ and $\overline{Sim}$ represent the mean values of Obs and Sim, respectively, and $n$ is the sample size.

(2) Negative values

Negative values are defined as the issue that arises when the BCC method is used to close the water budget, and the redistributed ΔRes error exceeds the actual values of budget components (P, ET, Q, and TWSC), causing P, ET, and Q to become negative. For TWSC, a negative value occurs when the corrected TWSC has an opposite sign to its raw value. These negative values represent only a subset of the errors introduced during water budget closure but reflect an extreme case of unreasonable ΔRes error redistribution, serving as an indicator of the BCC method's effectiveness.

When a budget component exhibits a negative value, the redistribution of ΔRes errors to other components is significantly affected, reducing the overall accuracy of the corrected datasets. Thus, negative values are a critical factor influencing the performance of existing BCC methods and should be prioritized for improvement. To better understand this issue, we analyze the proportion of negative values for each



budget component, their seasonal distribution, and their sensitivity to climatic
conditions (i.e., their prevalence in arid versus humid basins). Insights from this
analysis were incorporated into the proposed IWE-Res method to address the
occurrence of negative values (Section 3.4).
(3) Ensembled error of four budget components
The aforementioned evaluations (1) and (2) assess errors for individual budget
components. To gain a more comprehensive understanding of the uncertainties
introduced by water budget closure, we also evaluate the combined error. First, the
absolute error (AE) of each budget component is calculated (using P as an example,
see Equation 29). Second, the relative absolute error (RAE) is determined for each
budget component (Equation 28). Finally, by aggregating the relative errors of
individual components, we define the ensembled relative error (Equation 27) to
quantify the overall error introduced by BCC methods.
$$F(Re) = \frac{1}{n}\sum_{i=1}^{n} \frac{|AE(P')|-|AE(P_{Raw})|+|AE(ET')|-|AE(ET_{Raw})|+|AE(Q')|-|AE(Q_{Raw})|+|AE(TWSC')|-|AE(TWSC_{Raw})|}{P_0+ET_0+Q_0+|TWSC_0|}$$
$$= \frac{1}{n}\sum_{i=1}^{n} \frac{RAE(P)+RAE(ET)+RAE(Q)+RAE(TWSC)}{P_0+ET_0+Q_0+|TWSC_0|}$$

(27)

$$RAE(P) = |AE(P')| - |AE(P_{Raw})| \qquad (28)$$

$$AE(P) = |P - P_0| \qquad (29)$$

where, $F(Re)$ represents the ensembled relative error, and RAE refers to the relative
value of absolute error, with $i$ denoting the month. The subscript "Raw" corresponds
to the raw data of the budget components, the subscript 0 represents the observed data,
the superscript "′" denotes the BCC-corrected data for the budget components. The
degree of alteration induced by the BCC methods for each budget component are



defined based on the value of $F(Re)$, and four intervals are established in 5%
increments: no significant change [0-5%], minor change (5-10%], moderate change
(10-15%], and significant change (>15%).

*3.4 Proposed IWE-Res method for closing water budget*

In this section, the IWE-Res method is proposed to identify the optimal balance

for redistributing ΔRes, minimizing the sum of the introduced error to budget
components and the remaining ΔRes error while reducing the negative values
introduced by closing the water budget. The principle of the proposed IWE-Res
method involves gradually redistributing portions of ΔRes to budget components in
fixed percentage increments using existing BCC methods until an optimal balance is
achieved. This balance minimizes the combined error resulting from the introduced
error to budget components and the remaining ΔRes error. Gradual redistribution of
ΔRes begins at 0%, with iterations designed to incrementally redistribute portions of
ΔRes to the budget components. If negative values are identified in budget
components during the iterations, further redistribution to the affected budget
component will be suspended. Instead, in subsequent iterations, the remaining ΔRes
error will be redistributed among the other budget components. The specific steps of
the proposed IWE-Res method are as follows:

First, the ΔRes error is calculated using Equation 5 and the original datasets of

budget components.

Second, an iterative loop is constructed to compute the errors introduced into

budget components during the gradual redistribution of the ΔRes error and to address





negative values. To more accurately identify the optimal balance, a step size of 0.1%
of $\Delta$Res is used in each iteration in this study. We denote the $\Delta$Res redistributed to
budget components in each iteration as x, where x$\in$[0, $\Delta$Res].

During each redistribution of $\Delta$Res, two error terms are computed: (1) the

remaining $\Delta$Res error, defined as $\Delta$Res* = $\Delta$Res − x, and (2) the error introduced to
budget components due to the redistribution of the x error, denoted as IWE (Equation
31). When these errors are plotted in a coordinate system, two distinct curves emerge
(Fig. 2), each representing a different error relationship. For $\Delta$Res* (Equation 30),
Figure 2 shows a fixed, monotonically decreasing linear trend, as 0.1% increments of
$\Delta$Res are uniformly redistributed to budget components using existing BCC methods.
In contrast, the IWE curve exhibits a non-fixed shape, reflecting the cumulative error
introduced to budget components during the redistribution of a portion of $\Delta$Res
(Equations 31–32). This variability in the IWE curve arises from the nonlinear
relationship between the introduced budget component errors and the reduction in
$\Delta$Res error.
$$\Delta Res^* = ax + b = -x + \Delta Res \tag{30}$$
$$IWE = F(\varepsilon_P, \varepsilon_{ET}, \varepsilon_Q, \varepsilon_{TWSC}) = F(x, RAE) \tag{31}$$
$$RAE = \frac{1}{4}\sum_{i=1}^{4}(RAE(P) + RAE(ET) + RAE(Q) + RAE(TWSC)) \tag{32}$$
where $x$ represents the portion of $\Delta$Res redistributed to the budget components, with a
range from 0 to $\Delta$Res. The terms $\varepsilon_P$, $\varepsilon_{ET}$, $\varepsilon_Q$, $\varepsilon_{TWSC}$ represent the errors introduced to P,
ET, Q and TWSC, respectively, due to the redistribution of $x$ to the budget
components. F($x$, RAE) denotes the RAE error calculated by the redistribution of the $x$





error to budget components.
If negative values occur in certain budget components during the iterative
process, the redistribution of water imbalance error to the budget component with
negative values will be halted. In subsequent iterations, the weights for redistributing
the water imbalance error will be recalculated, ensuring that the remaining budget
components with positive values receive the redistributed water imbalance error. For
instance, if one of the four budget components (P, ET, Q, and TWSC) produces a
negative value—such as ET in a given iteration—the imbalance error for that iteration
will be redistributed to P, Q, and TWSC according to Equation 33.
$$F_i = X_i - x(G_i)(\frac{|\varepsilon_i|}{\sum_{j=1}^{n}|\varepsilon_j|})$$
(33)

where $F_i$ denotes the corrected dataset, and $X_i$ denotes the original dataset of budget
components. Since ET does not participate in the redistribution of the residual error x
based on the example above, the weighting vector is defined as G=[1, 0, −1, −1]. The
term ε represents the error in budget components estimated using existing BCC
methods, as described in Section 3.2.
Third, the IWE-Res curve is plotted (Fig. 2) to provide an intuitive comparison
between the introduced budget component errors and the remaining water imbalance
error. The error calculation results from Equations (30) and (31) are presented within
the same coordinate system.
The IWE-Res method is illustrated in Fig. 2 using four curves. The x-axis
represents the percentage of water imbalance error redistributed to budget components
using existing BCC methods, while the y-axis denotes the percentage of the remaining





water imbalance error ($\Delta Res^*$) after each iteration. The black dashed line represents
the redistributed x-error value among the budget components. The thin blue solid line
represents the $\Delta Res^*$ error curve. Since the sum of redistributed $x$ and remaining
$\Delta Res^*$ equals the total $\Delta Res$ error, this curve forms a monotonically decreasing 45°
line. The thin green solid line represents the introduced budget component error (IWE)
after a given percentage of $\Delta Res$ is redistributed (x-axis), with its shape varying
depending on the redistribution process (Fig. 2 is illustrative). Initially, when no $\Delta Res$
is redistributed (x = 0), the IWE error is zero. As more $\Delta Res$ is redistributed (with
increasing x values), IWE increases due to the growing uncertainty introduced. The
thin red solid line represents the total error, defined as the sum of $\Delta Res^*$ and IWE
after applying BCC methods. This curve varies depending on the redistribution
process, and its minimum value identifies the optimal balance where combined $\Delta Res^*$
and IWE errors are minimized. The intersection of the $\Delta Res^*$ and IWE curves
indicates only the point at which these errors are equal, not necessarily the optimal
balance.

To determine the optimal redistribution of the water imbalance error, we plot

the IWE-Res curve (the green solid line) for each basin, identifying the minimum of
the red total error curve. We then analyze its patterns across basins with different
characteristics to optimize water budget closure and improve the accuracy of budget
component datasets.

The IWE error in Fig. 2 also serves as a metric for evaluating the performance

of existing BCC methods. If a BCC method perfectly redistributed $\Delta Res$ without



introducing additional errors, the IWE curve would be a flat line at zero, and the red
total error line would coincide with the blue ΔRes* error line. This scenario indicates
that full redistribution of water imbalance error achieves the optimal balance,
providing indirect validation of the IWE-Res method's effectiveness.
Finally, the optimal balance is identified, enabling the generation of a
high-precision dataset that improves water budget closure. The optimal balance
corresponds to the minimum of the total error curve (IWE + ΔRes*), where the sum
of remaining water imbalance error and introduced budget component errors is
minimized. Ideally, both ΔRes* and IWE would reach their minimum values
simultaneously, meaning minimal error is introduced while fully redistributing ΔRes.
However, since this ideal state may not always be achievable, identifying the point
where combined error is minimized is essential. This principle defines the proposed
IWE-Res method (Fig. 2).

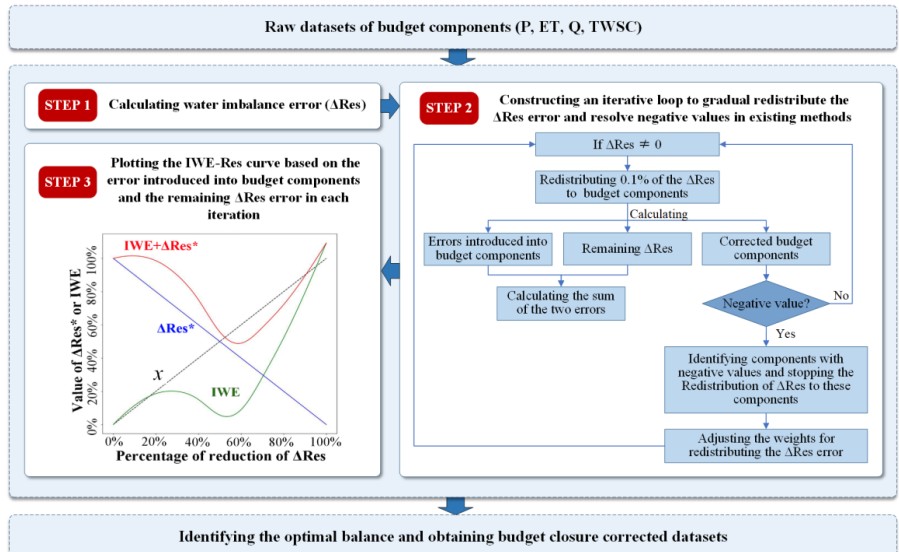


**Fig. 2** Framework of the IWE-Res method to identify the optimal balance for



redistributing the ΔRes error. The x-axis represents the proportion of ΔRes
redistributed to budget components, while the y-axis reflects the proportion of the
remaining ΔRes error. The black dashed line represents the redistributed x-error value
among the budget components. The blue solid line represents the ΔRes* curve, while
the green solid line shows the IWE error introduced into budget components after
redistributing the corresponding percentage of ΔRes. The red solid line represents the
total error curve.

**4 Results**
*4.1 Water imbalance error*
This section presents a comparative analysis of variations in water imbalance
errors across different basins and data combinations, aiming to clarify how errors in
budget components contribute to these discrepancies. Figure 3 illustrates the spatial
distribution of monthly ΔRes errors across various data combinations. To prevent the
cancellation of positive and negative values, the absolute values of monthly ΔRes
errors were first computed for each basin and then averaged.
As shown in Figure 3, ΔRes values vary significantly across basins. Most basins
in Africa, South America, and Europe exhibit high ΔRes values, typically exceeding
20 mm. In North America, ΔRes values generally range from 15 to 45 mm. Due to
inconsistencies among budget component datasets, substantial differences in ΔRes
also emerge across different data combinations. In combinations where only P data
varied while other budget component datasets remained constant (combinations in Fig.



3 where the first digit varies while the second and third remain constant), pronounced
changes in water imbalance errors were observed in parts of southern Africa, northern
Asia, and North America. This suggests substantial estimation errors in P for these
regions.

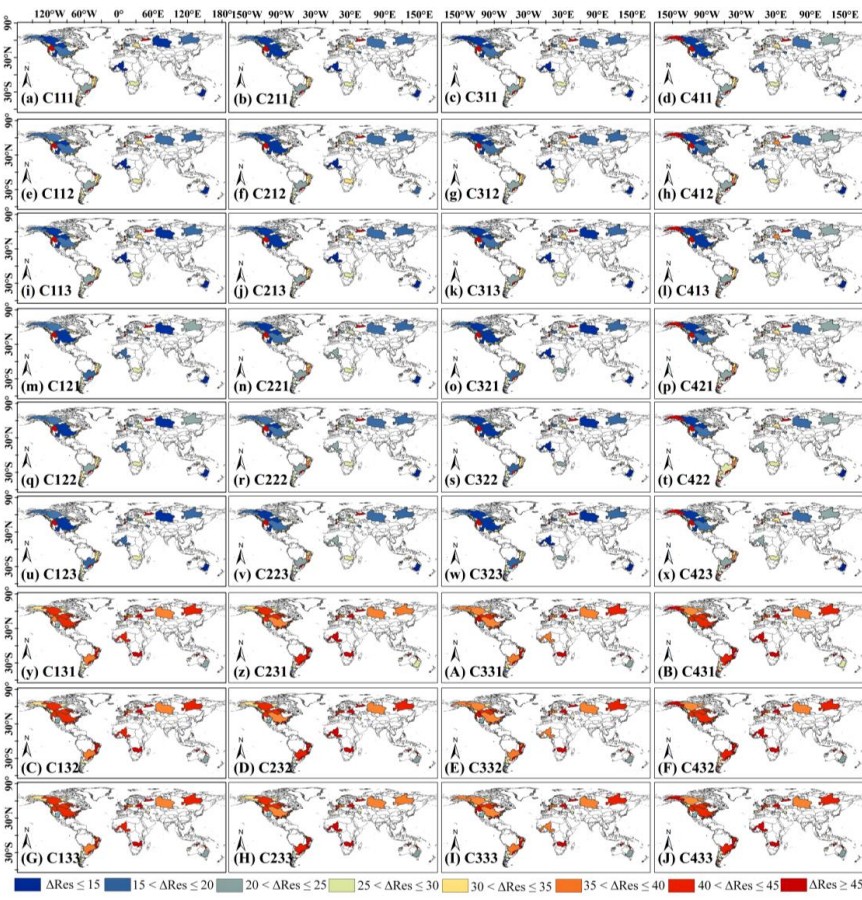


**Fig. 3** Spatial distribution of the ΔRes error on a monthly scale for different
combinations of budget components. The unit of ΔRes is mm. Each subplot represents
a distinct combination, where the first digit corresponds to the P product, the second
to the ET product, and the third to the TWSC product. The detailed definitions of





these combinations are provided in Equation 3.

When different ET products were used (combinations where the second digit

varies while the first and third remain constant), water imbalance errors changed
significantly in most basins. Specifically, in combinations using the TerraClimate ET
dataset, water imbalance errors exceeded 35 mm in the majority of basins, indicating
severe water imbalance. This underscores the considerable discrepancies among ET
products and their substantial impact on accurately representing basin water balance.
In contrast, when TWSC data from different GRACE products were used
(combinations where the third digit varies while the first and second remain constant),
variations in water imbalance errors across basins were relatively small.

Overall, ET and P are the primary variables influencing water imbalance in most

basins, consistent with previous findings (Pan et al., 2012; Zhang et al., 2018). The
uncertainty in budget component datasets remains a key challenge for water balance
research (Dagan et al., 2019; Lv et al., 2017; Luo et al., 2023).
*4.2 Uncertainties of budget components introduced by closing water budget*

To gain a more comprehensive understanding of the uncertainties introduced into

budget components when closing the water budget, this section analyzes the errors
introduced by fully closing the water budget using existing BCC methods from three
perspectives: the errors of individual budget components, the occurrence of negative
values, and ensemble errors (Section 3.3).
4.2.1 Errors of individual budget components



Figure 4 presents the relative statistical metrics calculated using Equations 19–22
to evaluate the uncertainties introduced into budget components by existing BCC
methods. Positive values indicate an improvement in the accuracy of corrected budget
components, whereas negative values indicate a decline in accuracy.
Overall, existing BCC methods exhibit notable limitations in enhancing the
accuracy of budget components. In particular, for P, nearly all statistical metrics (CC',
NSE', MAE', RMSE') across various basins yield negative values. For instance, under
the CKF method, these values are approximately –0.05, –0.15, –3.82 mm, and –8.47
mm, respectively, indicating a significant reduction in the accuracy of the corrected P
dataset when BCC methods are applied to enforce water budget closure. Specifically,
the accuracy of the corrected P dataset decreases by approximately 6%, 34%, 11%,
and 55%, as reflected in the CC, NSE, MAE, and RMSE metrics, respectively.
Analysis of 13 selected basins with sufficient P observations further confirms this
decline, showing a reduction in the accuracy of budget-corrected P (Figure 5). A
possible explanation for this decrease is the inherently high accuracy of raw P datasets,
supported by advancements in remote sensing technologies, meteorological models,
and observational networks. However, when BCC methods are applied, water
imbalance errors from other budget components, such as ET, may be inappropriately
redistributed to the corrected P dataset in an effort to enforce overall water budget
closure. As a result, while the total water budget is balanced, the accuracy of the
corrected P data is compromised.

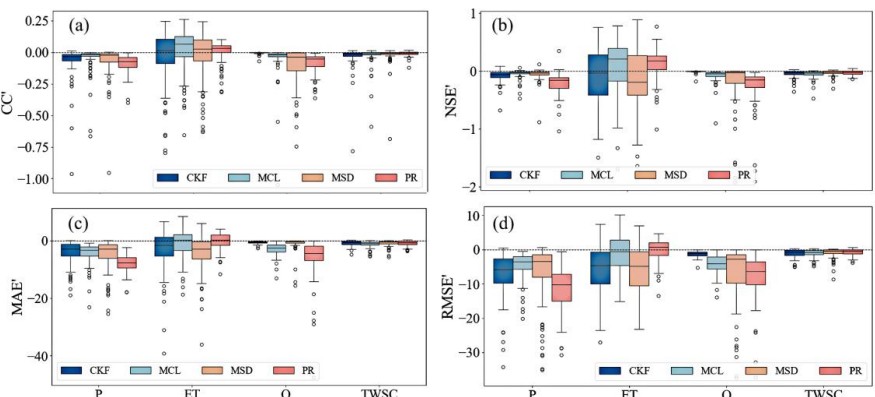


**Fig. 4** Box plot quantifying the errors introduced into budget components by existing

BCC methods when closing the water budget. (a) - (d) represent the results of the CC',

NSE', MAE', RMSE' indicators, respectively. Positive values indicate an improvement

in accuracy relative to the reference values after applying existing BCC methods,

while negative values indicate a decline. Different colors represent different BCC

methods.

The impact of enforcing water budget closure using existing BCC methods on

ET was particularly significant (Fig. 4), with approximately 50% of basins exhibiting

improved accuracy in corrected ET. For TWSC, most basins showed decreased

accuracy. For Q, CC' and NSE' values ranged from 0 to -0.5, while MAE' and RMSE'

were primarily concentrated between 0 mm and -20 mm. Consequently, the accuracy

of corrected Q declined, with CC, NSE, MAE, and RMSE decreasing by

approximately 0.1, 0.2, 3 mm, and 5 mm, respectively. These findings indicate that

while redistributing the entire $\Delta Res$ enhances the consistency of budget components,

it provides limited improvement in their accuracy and may even introduce further





errors. Identifying an optimal redistribution strategy for ΔRes errors could help
mitigate this issue.

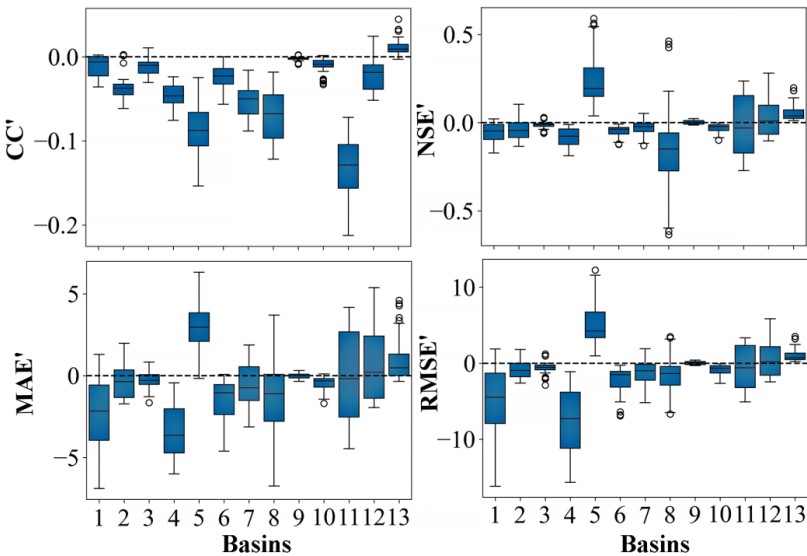


**Fig. 5** Box plot illustrating precipitation errors introduced by correcting ΔRes using
existing BCC methods across 13 basins with sufficient observational precipitation
data. The x-axis represents the 13 basins in the following order: NIGER, OB,
MISSISSIPPI, SACRAMENTO, SAN JOAQUIN, SUSQUEHANNA, BRAZOS,
FRASER, NELSON, MURRAY, RIO EBRO, ELBE, and KURA.

4.2.2 Negative values
This section examines the occurrence of negative values in budget components
arising from the application of existing BCC methods to close the water budget. For
each budget component, the proportion of months with negative values relative to the
total time series was computed (Fig. 6). Overall, the fraction of negative values across
budget components ranges from 0% to 10%, with the majority falling below 5%. This





proportion is notable, as negative values indicate substantial inaccuracies in the
redistribution of water imbalance errors by existing BCC methods. When a budget
component exhibits a negative value, the accuracy of the remaining budget
components is also compromised. The relatively high occurrence of negative values
highlights the need for methodological improvements to enhance the performance of
existing BCC methods.

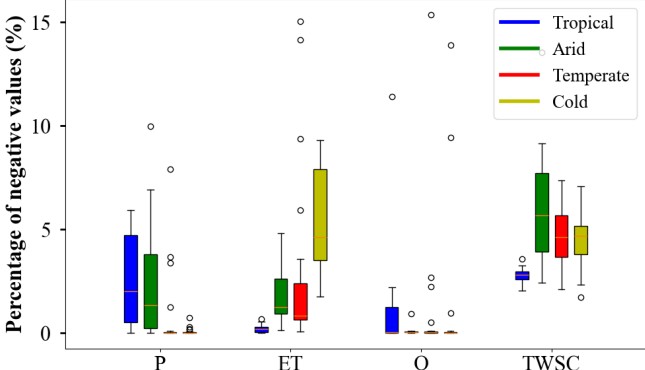


**Fig. 6** Percentage of negative values for corrected datasets of budget components
induced by closing the water budget. Different colors indicate distinct climate
classifications.

Among the individual budget components, ET and TWSC exhibit the most

pronounced negative values, followed by P, while Q shows the least (Fig. 6). Notably,
the proportion of negative values in budget components varies significantly across
climate types. For P, negative values generally remain below 5% but can occasionally
reach 7% in arid regions. The likelihood of negative P values is higher in tropical and
arid climates (mostly below 5%) compared with temperate and cold regions (around



1%). For ET, the proportion of negative values is largely below 5%, but it is notably
higher in cold climates (reaching 9%), followed by arid and temperate regions
(approximately 1%–3%). Tropical climates exhibit the lowest proportion of negative
ET values, with most instances below 1%. Q consistently shows a low occurrence of
negative values across all climate types (generally below 3%), with a slightly higher
probability in tropical regions than in other zones. The proportion of negative TWSC
values ranges from 3% to 10%, being lowest in tropical climates (below 5%), while
other climate types exhibit values between 3% and 10%.
Fig. 7 presents the seasonal cycle of negative values across different climate
zones, examining whether these values exhibit significant seasonal patterns. Negative
P values predominantly occur in summer and autumn, with a higher proportion from
June to September in tropical climates compared to arid regions. ET tends to show
negative values more frequently in winter and spring, with a lower likelihood in
summer and autumn. Except in summer, cold climate zones are most susceptible to
negative ET values. Among the four budget components, Q has the lowest occurrence
of negative values. Negative TWSC values are primarily observed in spring, autumn,
and December, with arid regions exhibiting a higher likelihood of negative values
throughout the year compared to other climate types. These findings indicate that the
occurrence of negative values varies significantly across seasons and climate zones.
Future research should account for this seasonal variability to further refine existing
BCC methods.



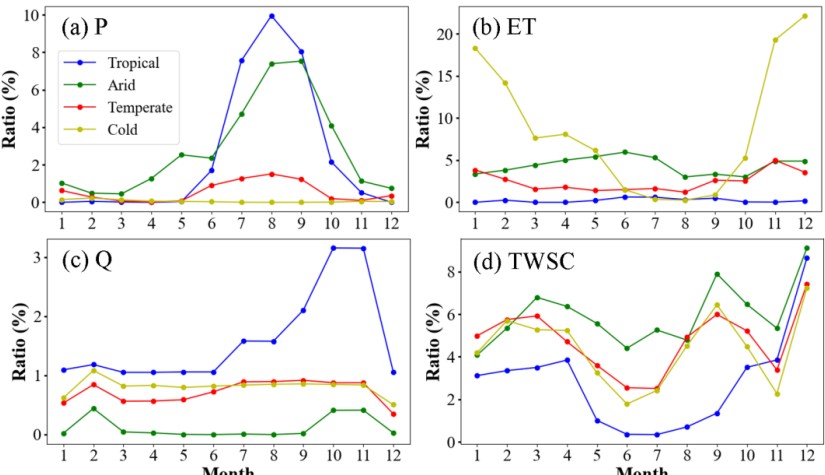


**Fig. 7** Seasonal cycle of the proportion of negative errors for budget components.

Different colors representing various climate types.

### 4.2.3 Ensemble errors

Fig. 8 presents the ensemble errors in budget components (i.e., F(Re) in Equation
27) introduced by existing BCC methods (CKF, MCL, MSD, and PR). All four
methods exhibit similar spatial distribution patterns. Notably, high ensemble errors
(F(Re) > 10%) are concentrated in the northwestern basins of North America,
particularly in Alaska, suggesting substantial variations in budget components in these
regions. Basins with minor ensemble errors (5% < F(Re) ≤ 10%) generally cover
larger areas, such as African and Northern Asia. Although these errors are relatively
small, they remain non-negligible. Basins with lower ensemble errors (F(Re) ≤ 5%)
also cover some basins. Further analysis of ΔRes in basins with higher F(Re) values
reveals a strong correlation, as these basins also exhibit larger ΔRes. This finding
highlights the limitations of existing BCC methods in effectively redistributing ΔRes



errors.

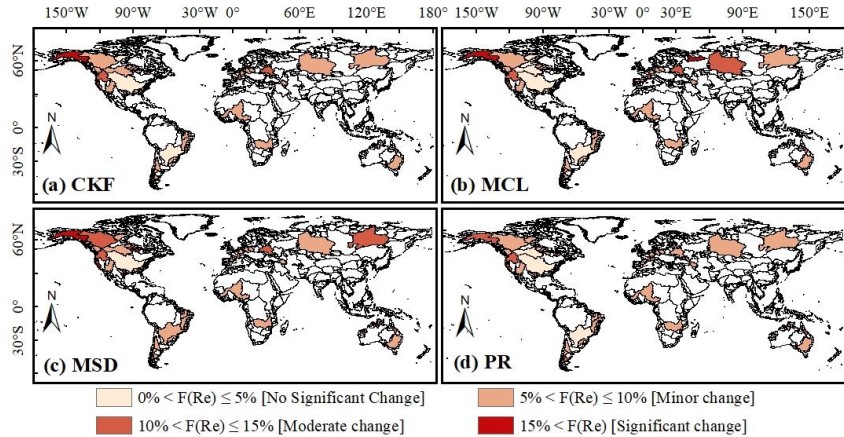


**Fig. 8** Ensemble errors in budget components introduced by closing the water budget
using existing BCC methods.


To determine whether the error cost introduced by existing BCC methods in
closing the water budget outweighs the reduction in water imbalance error, we
analyzed the relationship between the reduction in ΔRes error and the introduced
budget component errors (Fig. 9). As shown in Fig. 9, with the exception of the PR
method, the basins where |RAE| exceeds |Res| are largely consistent across the other
three BCC methods. This discrepancy arises because the PR method redistributes
ΔRes based on the magnitude of budget components, whereas the CKF, MCL, and
MSD methods redistribute ΔRes according to the estimated errors in budget
components.





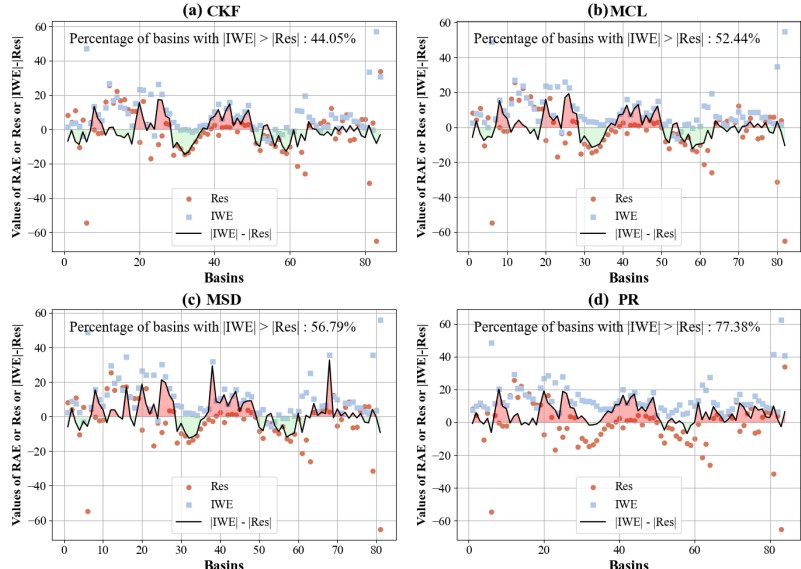


**Fig. 9** Comparison of relative absolute error (RAE) and residual error (Res) for four

BCC methods (CKF, MCL, MSD, PR) across various basins. The black lines in the

red shaded area on the upper half of the y-axis indicate that the error introduced by the

BCC methods for budget components exceeds the reduction in ΔRes error (|RAE| >

|Res|), while the green shaded area on the lower half of the y-axis represents cases

where the error introduced is less than the reduction in ΔRes error (|RAE| < |Res|).

For the CKF, MCL, MSD, and PR methods, the proportions of basins where

|RAE| exceeds |Res| are 44.05%, 52.44%, 56.79%, and 77.38%, respectively. This

indicates that, for all four methods, the introduced |RAE| error in budget components

surpasses the reduction in water imbalance error in more than 40% of the basins.

These findings underscore the non-negligible uncertainties introduced by these

methods. Striking a balance between reducing water imbalance error and minimizing





the impact of budget component errors remains a critical challenge, motivating us to
propose the IWE-Res method to identify optimal balance.
*4.3 Verifying the accuracy of the proposed IWE-Res method*

Based on the error analysis of existing BCC methods in Section 4.2, this section

assesses the accuracy and reliability of the proposed IWE-Res method. The evaluation
is conducted through a comparative analysis with PR, CKF, MCL, and MSD, focusing
on three key aspects: the errors of individual budget components, the occurrence of
negative values, and ensemble errors.

Fig. 10 compares the accuracy of the proposed IWE-Res method with existing

PR, CKF, MCL, and MSD methods from the perspective of errors in individual
budget components. The red and blue lines represent the IWE-Res method and the
existing BCC methods, respectively, while the bars indicate the relative accuracy
improvement of the IWE-Res method compared to the BCC methods. As shown in
Fig. 10, the proposed IWE-Res method exhibits consistently higher accuracy than all
existing CKF, MCL, MSD, and PR methods for budget components P, ET, Q, and
TWSC. This result highlights the superior capability of the IWE-Res method in
optimizing errors in budget corrected datasets. According to the statistical metrics CC,
NSE, MAE, and RMSE, the proposed IWE-Res method improves the corrected P data
by 4.2%, 21.3%, 25.5% and 29.5%, respectively, compared to the existing BCC
methods. For corrected ET, the improvements are 6.9%, 265.7%, 17.6% and 24.7%,
respectively; for corrected Q, the improvements are 3.4%, 185.1%, 67.1%, and 69.0%;
and for corrected TWSC, the improvements are 0.0%, 7.0%, 7.5%, and 6.8%.



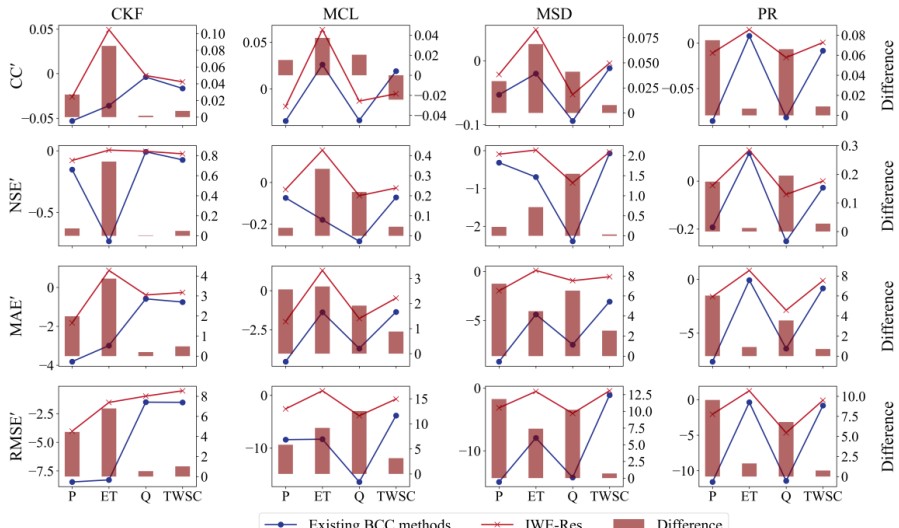

**Fig. 10** Performance comparison of the proposed IWE-Res method with existing BCC methods in corrected individual budget components. The red and blue lines in the figure represent the average values across all basins considered in this study.

Table 2 presents the percentage of negative values observed in the corrected budget components for the proposed IWE-Res method and existing BCC methods. One of the key contributions of the IWE-Res method is its ability to address the critical limitation of negative value generation in existing BCC methods. As a result, the percentage of negative values in the corrected P, ET, Q, and TWSC data using the proposed IWE-Res method is zero. In contrast, the corrected P, ET, Q, and TWSC data obtained from existing BCC methods contain negative values to varying degrees (for a detailed analysis of negative values, see Section 4.2.2). These results demonstrate that, in addition to improving the accuracy of budget components relative to observations, the proposed IWE-Res method effectively eliminates the issue of




negative values inherent in existing BCC methods.
**Table 2** The percentage of months with negative values in the corrected datasets of
budget components P, ET, Q, and TWSC for the proposed IWE-Res method and
existing BCC methods. The percentages in the table represent the average values
across all basins considered in this study.

|  | **P** | | **ET** | | **Q** | | **TWSC** | |
|---|---|---|---|---|---|---|---|---|
|  | Existing | IWE-Res | Existing | IWE-Res | Existing | IWE-Res | Existing | IWE-Res |
| CKF | 0.31% | 0% | 6.73% | 0% | 0.75% | 0% | 4.81% | 0% |
| MCL | 1.82% | 0% | 4.78% | 0% | 0.77% | 0% | 7.61% | 0% |
| MSD | 1.68% | 0% | 7.03% | 0% | 0.82% | 0% | 5.40% | 0% |
| PR | 0% | 0% | 0.57% | 0% | 0.72% | 0% | 0.47% | 0% |


We further evaluate the accuracy and reliability of the proposed IWE-Res
method using the ensemble error metric defined by Equation (27) (Fig. 11), where
lower values indicate better model performance. As shown in Fig. 11, the IWE-Res
method significantly reduces ensemble errors compared to existing BCC methods. For
instance, in the CKF method, the median ensemble error decreases from above 5% to
below 5%. This reduction is even more pronounced in the MCL, MSD, and PR
methods. Additionally, the interquartile ranges under IWE-Res are notably narrower,
suggesting improved control over stochastic variability. For example, in the PR
method, the interquartile range shrinks from 5–8% (existing BCC methods) to 1–2%
(IWE-Res), reflecting an approximate 67% reduction in variability. These findings
highlight the robustness of the IWE-Res method in minimizing integrated errors,
aligning with its previously demonstrated excellence in single-variable error
optimization and negative value elimination.





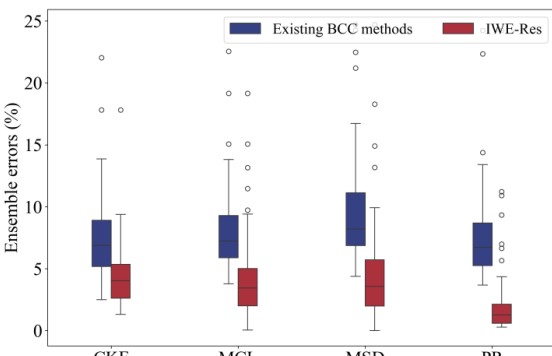


**Fig. 11** Performance comparison of the proposed IWE-Res method and existing BCC

methods based on the ensemble errors of budget components.


*4.4 Identifying the optimal balance for redistributing water imbalance error*

Based on the proposed IWE-Res method, this section aims to determine the

optimal balance for redistributing water imbalance errors across different climate

zones (Tropical, Arid, Temperate, and Cold climate zones) to achieve the best

trade-off (Figs. 12-15). Specifically, it seeks to minimize both water imbalance errors

and the uncertainties in budget components introduced by enforcing water budget

closure. The findings offer a valuable reference for generating high-precision datasets

of budget components with a closed water budget in diverse climate regions. When

developing the IWE-Res method, we incorporated multiple BCC methods, each based

on different principles. As a result, the identified optimal balance results vary across

methods. This section presents results for the CKF method only, while results for the

MSD, MCL, and PR methods are provided in the supplementary materials.

Overall, the optimal balance varied among basins located in different climate

zones (Figs. 12–15). In most basins within the Tropical, Arid, and Temperate zones,



the optimal balance was achieved when only a portion of the water imbalance error,
rather than the entire error, was redistributed to budget components. However, this
pattern was not observed in the Cold region.

For most basins in the Tropical climate zone (Fig. 12), the optimal balance was

reached when 40%–90% of ΔRes was reallocated to budget components, suggesting
that the corrected budget datasets achieve their highest accuracy within this range.
Notably, approximately 20% of basins attained their optimal balance when 80%–90%
of ΔRes was redistributed, while about 70% did so within the 40%–50% range.
Therefore, in Tropical basins, if sufficient observational data are unavailable to
precisely determine the optimal balance, redistributing 40%–50% of ΔRes to budget
components is recommended to obtain the most accurate dataset.

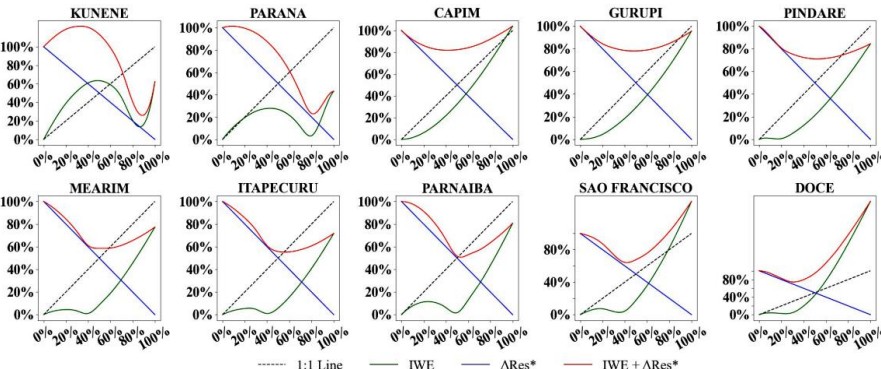


**Fig. 12** IWE-Res curve in basins of Tropical climate zone for identifying the optimal
balance that enhances water budget closure and reduces uncertainty.

For basins in the Arid climate zone (Fig. 13), optimal balance are generally found

when 40%–90% of ΔRes is redistributed, indicating that the corrected budget



component datasets achieve their highest accuracy within this range. Specifically,
approximately 31% of basins reach their optimal balance at 40%–50% redistribution,
38% at 60%–80%, and over 20% at 90%. Thus, the distribution of optimal balance in
Arid basins does not follow a distinct pattern.

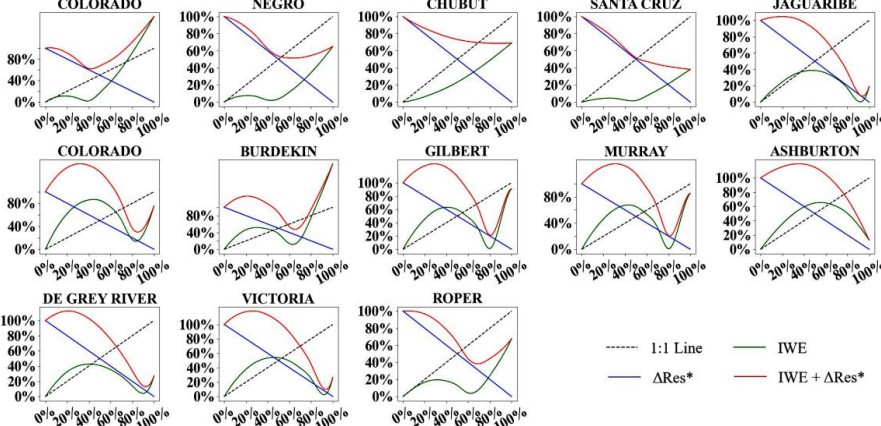


**Fig. 13** IWE-Res curve in watersheds of Arid climate zone for identifying the optimal
balance that enhances water budget closure and reduces uncertainty.

In the Temperate climate zone (Fig. 14), optimal balance are concentrated within
the 40%–90% range. Approximately 53% of basins achieve their optimal balance
when 40%–50% of ΔRes is redistributed, while 17% and 13% reach it at 70% and 90%
of the ΔRes redistribution. A smaller proportion of basins achieve optimal balance at
60% and 80% of the ΔRes redistribution. Overall, redistributing 40%–50% of ΔRes
minimizes the combined error from both the introduced budget component error and
the remaining water imbalance error in most basins.





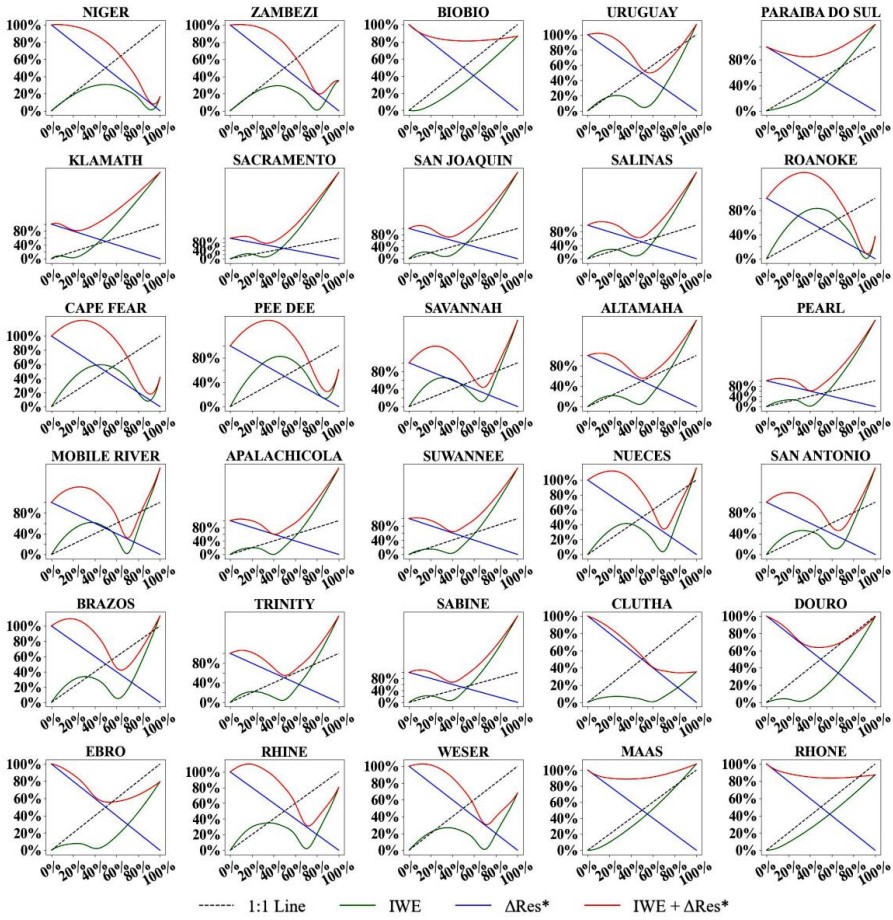


**Fig. 14** IWE-Res curve in watersheds of Temperate climate zone for identifying the

optimal balance that enhances water budget closure and reduces uncertainty.


In Cold climate zone basins (Fig. 15), the optimal balance is typically reached

when the entire ΔRes is fully redistributed. This suggests that complete redistribution

of ΔRes does not compromise the accuracy of the budget components. This is

primarily due to the trend observed in the IWE curve, which initially

increases—indicating rising error—before decreasing, in contrast to the patterns seen



in most basins in Figs. 12–14. A comparison of ΔRes and the negative values
introduced by full redistribution of ΔRes across climate zones reveals that, in Cold
regions, negative values predominantly occur in ET. This is likely due to the
inherently lower ET values in Cold regions, which increases the likelihood of negative
values when ΔRes is redistributed. However, errors introduced in other budget
components, such as P and Q, remain relatively low under full redistribution of ΔRes.

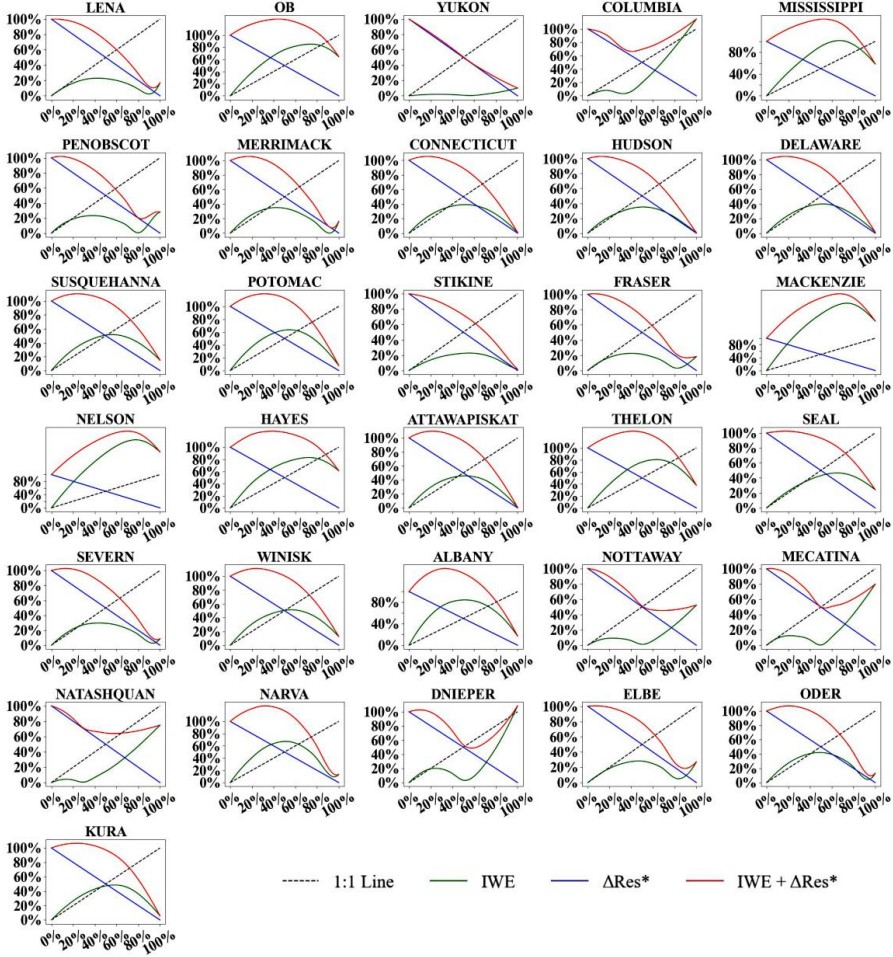


**Fig. 15** IWE-Res curve in watersheds of Cold climate zone for identifying the optimal
balance that enhances water budget closure and reduces uncertainty.





**5 Discussion**

*5.1 Uncertainty introduced by existing BCC methods*

To quantify the uncertainty introduced by existing BCC methods in closing the water balance, we evaluated four BCC methods across 84 global basins. The assessment focused on errors in individual budget components, occurrences of negative values, and ensemble errors in budget components. Our findings indicate that while existing BCC methods improve the consistency of budget components, their ability to enhance their accuracy is limited and, in some cases, may even reduce it.

Several factors may contribute to this reduction in accuracy. First, existing BCC methods do not incorporate observational data for budget components; instead, they redistribute water imbalance errors based on estimated errors in each budget component. If these error estimates are inaccurate, additional uncertainty may be introduced. Therefore, incorporating more observational data on budget components represents a key strategy for mitigating this issue. Second, existing BCC methods, including the proposed IWE-Res method, do not account for the physical mechanisms of the water cycle, relying instead on statistical approaches to enforce water budget closure. As a result, the corrected budget components may deviate from actual conditions. Future research should integrate BCC methods with physically based hydrological models to improve the physical consistency of budget component datasets. Third, observational datasets themselves do not fully satisfy water budget closure. Consequently, even if a corrected dataset achieves complete closure using existing methods, this does not necessarily mean it aligns more closely with



observational data. In practice, corrected datasets may approach the true values, but
without direct access to these true values, evaluating their accuracy remains
challenging. Developing more objective methods for assessing the accuracy of water
budget closure-corrected datasets will be an important focus for future studies,
although this lies beyond the scope of the present study.
*5.2 Identification of the optimal balance*

Each budget component inherently contains observational or model-based errors.

Indiscriminately redistributing water imbalance errors across all budget components
to achieve complete water budget closure can introduce additional uncertainties. By
identifying the optimal balance for error redistribution across different climate zones,
we observed significant variations in distribution patterns. In tropical and temperate
regions, most basins achieved their optimal balance when 40%–90% of the water
imbalance error was redistributed, with a concentration around the 40%–50% range.
In arid regions, the distribution of optimal balance was more dispersed, lacking a clear
concentration within any specific redistribution range but generally falling within the
40%–90% range. Cold climate regions exhibited distinct characteristics, with most
basins achieving the smallest error when the water imbalance error was fully
redistributed.

Overall, determining an appropriate redistribution ratio for water imbalance

errors effectively improves budget component accuracy. Future research should focus
on the underlying rationale for error redistribution, as the sensitivity of different
budget components to water imbalance errors varies. Excessive or insufficient



redistribution to certain components can lead to imbalances in the remaining
components, ultimately failing to accurately represent actual hydrological processes.
**6 Conclusions**
Existing BCC methods introduce new uncertainties when closing the water
budget due to challenges in accurately estimating errors in budget components and the
integrated concept of water imbalance error. This study first evaluates the issues
arising from existing BCC methods by comparing the errors introduced in budget
components with the improvement in water budget closure precision. A new method,
termed IWE-Res, is proposed to identify the optimal redistribution of ΔRes, aiming to
minimize the sum of the remaining residual error and the introduced budget
component error. To assess the reliability of the IWE-Res method, we compare it with
four different BCC methods across 84 basins spanning various global climate zones.
The main conclusions are as follows:
(1) While applying existing BCC methods reduces water imbalance error, it
simultaneously introduces new errors in budget components. For P, a decline in
accuracy is observed in most basins. For Q, the corrected data exhibits lower
performance than the raw data, with reductions in CC, NSE, MAE, and RMSE of
approximately 0.1, 0.2, 3 mm, and 5 mm, respectively. At the basin scale, more than
40% of basins experience budget component errors greater than the reduction in ΔRes
after applying existing BCC methods.
(2) The proportion of negative corrected values in each budget component is
predominantly within 0%–5%. For ET, negative corrected values are mostly below



5%, though they reach 9% in cold climate regions. For P, the proportion is primarily below 5%, with rare occurrences around 7%. Q generally exhibits a lower proportion of negative values, mostly below 3%. In TWSC, negative values are concentrated between 3% and 10%.

(3) The proposed IWE-Res method improves the accuracy of corrected budget components compared to existing BCC methods. Based on RMSE, it improves the accuracy of corrected P by 29.5%, corrected ET by 24.7%, corrected Q by 69.0%, and corrected TWSC by 6.8%.

(4) Except in cold regions, redistributing 40%–90% of ΔRes to budget components yields the optimal balance, minimizing the sum of the remaining ΔRes and the introduced budget component error. In tropical and temperate regions, the optimal balance is typically achieved when 40%–50% of ΔRes is redistributed. Similarly, in arid regions, redistributing 40%–90% of ΔRes effectively reduces errors, though the optimal redistribution ratio varies across basins. In most cold-region basins, the total error is minimized when the entire ΔRes is redistributed.

**Declaration of Competing Interest**

The authors declare that they have no known competing financial interests or personal relationships that could have appeared to influence the work reported in this paper.

**Acknowledgments**

This research was supported by the National Natural Science Foundation of China (No. 42201038) and China Scholarship Council.



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
