# Peer review of "A novel method for correcting water budget components and reducing their"

_EGUsphere, 2025_

## Author Comment (AC1)

Response to Review
Reviewer #1:

The manuscript is focused on numerical techniques to 'distribute' residuals of a water balance equation over the contributing terms, avoiding negative values. The manuscript seems focused on the numerical techniques, with limited efforts for a hydrological interpretation. It could be more convincing if authors bring a bit more on the 'explanation' side.

**Response:** Thank you for your valuable comment. We agree that the manuscript, in its current form, lacks sufficient hydrological interpretation regarding both the redistribution of residuals across water budget components and the prevention of negative values during this process. To address this concern, we revised the manuscript from the following aspects: (1) We provided a hydrological explanation for the existence of residuals in current water balance datasets and emphasized the importance of reducing these residuals; (2) We added a hydrological interpretation of the limitations associated with existing correction methods that fully redistribute the residuals in order to achieve water balance closure; (3) We added more hydrological interpretation to clarify the rationale and validity of the proposed method in this study; (4) Following one of your suggestions below, we conducted a comparison between the monthly-scale results of our proposed method and the annual-scale estimates derived from the physically based Budyko framework. Our specific modifications are as follows:

(1) We have added a hydrological explanation of the origins of residuals in existing water balance datasets and emphasized the importance of minimizing these residuals. The terrestrial water balance, composed of four major hydrological variables—precipitation (P), evapotranspiration (ET), streamflow (Q), and terrestrial water storage change (TWSC)—describes the distribution and movement of water across the Earth's land surface. Notably, these fluxes and storages are dynamically linked, reflecting the actual hydrological processes within a basin. For example, in real-world hydrology, precipitation transforms into ET, Q, and TWSC through physical processes such as infiltration, surface runoff, and evaporation. Therefore, improving the consistency (i.e., closure) among P, ET, Q, and TWSC is essential for advancing our understanding of watershed-scale hydrological dynamics.

However, the data used for water components, whether through direct observations or hydrological modeling, are often imbalanced. Observationally, each component is typically measured independently, and there is a lack of integrated observational systems which are capable of capturing all components simultaneously. From a modeling perspective, structural simplifications, parameter uncertainties, and observational errors in input data often propagate through the system, preventing budget closure.

Unfortunately, achieving a closed water balance remains challenging. In the era of big data, the rapid expansion of remote sensing and reanalysis datasets has significantly improved the availability of hydrological data. However, these datasets also suffer from non-closure in the water balance, mainly because they are independently produced and not physically coupled, making it difficult to reconcile their inconsistencies through modeling. This underscores the urgent need for data-driven approaches to enhance the internal consistency of these datasets—especially since the water cycle variables are inherently interconnected. To address this issue, numerous water budget closure correction methods have been proposed in existing studies.

We acknowledge that our initial manuscript did not provide sufficient hydrological explanation

regarding the origin of residuals or the significance of reducing them. To address this, we have added the following paragraph to the revised manuscript:

"The terrestrial water balance represents a fundamental physical framework that describes the distribution and movement of water across the Earth's land surface (Lehmann et al., 2022). It is governed by four interconnected components—precipitation (P), evapotranspiration (ET), streamflow (Q), and terrestrial water storage change (TWSC)—that together regulate the exchange of water among the atmosphere, land, and oceans (Abolafia-Rosenzweig et al., 2021; Sahoo et al., 2011; Chen et al., 2020; Wang et al., 2015). These components are dynamically linked and respond to climatic variability, land surface heterogeneity, and human interventions across a range of spatial and temporal scales. Achieving water budget closure (that is, ensuring internal consistency among these fluxes and storages, Equation 1) is essential for advancing our understanding of hydrological processes (Li et al., 2024; Mourad et al., 2024).

$$P - ET - Q - TWSC = 0 \qquad\qquad (1)$$

Despite its importance, obtaining observational datasets that achieve water balance closure remains a major challenge. In practice, no single observational system can simultaneously measure all four water balance components at the required resolution and accuracy. Each variable is typically derived from independent data sources with differing spatial and temporal characteristics, which complicates the direct closure of the terrestrial water budget.

P is typically derived from point-based rain gauge networks, which are generally reliable but often incomplete, requiring gap-filling (Esquivel-Arriaga et al., 2024; Nassaj et al., 2022; Bai et al., 2021; Lockhoff et al., 2014). The main source of uncertainty lies in the spatial distribution and representativeness of these gauges, particularly in relation to precipitation type (Bai et al., 2019; Trenberth et al., 2014). Spatial uncertainty tends to be low for widespread frontal systems but can be substantial for localized convective storms (Palharini et al., 2020). Gauge placement is often dictated by accessibility and logistical convenience, which may lead to underestimation of the uncertainty in daily precipitation inputs (Wang et al., 2017; Bai et al., 2019; Wu et al., 2018). Satellite-based precipitation estimates have demonstrated good performance in capturing frontal rainfall, but not in other rainfall types (Masunaga et al., 2019; Petković et al., 2017; Palharini et al., 2020). ET is commonly estimated using empirical or physically based models (Jacobs et al., 1998; McMahon et al., 2016; Allen et al., 1998). Although these models are generally well calibrated, uncertainties persist due to the complex influence of advection and localized meteorological variability, especially in small catchments. At larger spatial scales, energy balance approaches tend to provide sufficiently accurate estimates (Hua et al., 2020; Hao et al., 2018; Ruhoff et al., 2022). Q measurements typically exhibit low uncertainty when rating curves are well established and regularly maintained (Jian et al., 2015; Krabbenhoft et al., 2022). However, uncertainty can still arise from the delineation of watershed boundaries, particularly in regions where groundwater flow does not align with surface catchment divides (Huang et al., 2023; Bouaziz et al., 2018). This mismatch can result in misrepresentation of actual hydrological contributions. TWSC generally has a negligible impact on water balance calculations over multi-year periods, but can significantly affect short-term (e.g., daily) balances (He et al., 2023; Zhang et al., 2016). A key challenge is defining the effective depth over which TWSC should be quantified, as changes in soil moisture near the surface are more easily observed than those occurring at greater depths.

Hydrological models, which are grounded in the principle of mass conservation and explicitly implement the water balance equation, offer an alternative to direct observation for achieving water

budget closure. However, in practice, model structure simplifications, parameter uncertainties, and errors in meteorological forcing data introduce substantial biases and propagate uncertainty across simulated components. These limitations make it equally difficult to achieve water balance closure using hydrological modeling alone.

In recent years, the rapid expansion of remote sensing and reanalysis datasets has significantly improved global access to water cycle variables, offering new opportunities for data-driven analysis of hydrological processes.".

(2) Hydrological interpretation of the limitations associated with existing correction methods that fully redistribute the residuals.

Existing water budget closure correction (BCC) methods commonly redistribute the entire residual error ($\Delta$Res) among water budget components to enforce strict closure. In the revised manuscript, we explain the limitations of this full redistribution approach from two perspectives: the hydrological origins of $\Delta$Res and the principles underlying the redistribution process used in current BCC methods. $\Delta$Res is a composite error term that includes estimation errors in budget components, systematic biases, and the omission of unmeasured components. However, existing BCC methods typically assume that all of $\Delta$Res arises from estimation errors and then redistribute it according to estimation errors. This assumption oversimplifies the true hydrological causes of imbalance and can lead to unreasonable outcomes—most notably, the appearance of negative values in corrected datasets. We have added the following sentence to the revised manuscript:

"To address this issue, a growing number of studies have adopted water budget closure correction (BCC) methods to reduce water imbalance errors ($\Delta$Res), with the goal of forcing $\Delta$Res from a non-zero value ($\Delta$Res $\neq$ 0) to theoretical closure ($\Delta$Res = 0), where $\Delta$Res = P – ET – Q – TWSC (Zhou et al., 2024; Munier et al., 2018; Zhang et al., 2016). Common methods include Proportional Redistribution (PR), the Constrained Kalman Filter (CKF), Multiple Collocation (MCL), and the Minimized Series Deviation (MSD) method (Pan et al., 2012; Luo et al., 2023)." and

"Existing BCC methods redistribute the entire $\Delta$Res error among water budget components to enforce strict water budget closure. This redistribution is typically guided by the relative uncertainties of the individual components, based on the assumption that the entire residual error originates from observational or modeling errors in these datasets. However, this assumption overlooks the fact that $\Delta$Res is not solely the result of measurement or estimation errors in P, ET, Q, or TWSC. Rather, it is a composite residual that also reflects contributions from systematic biases and the omission of unmeasured components. These include deep groundwater exchanges that may cross basin boundaries, snow and glacier storage changes (particularly in high-altitude or high-latitude regions), and anthropogenic influences such as irrigation withdrawals, reservoir operations, and inter-basin water transfers. Because existing BCC methods do not explicitly account for these additional sources of imbalance, forcing strict closure by allocating the entire $\Delta$Res to the measured components can introduce unrealistic uncertainties. As a result, the application of existing BCC methods—despite their goal of improving internal consistency—often leads to limited improvements, or, in some cases, even a decline in the accuracy of the corrected hydrological datasets.

A clear manifestation of this limitation is the occurrence of negative values in corrected budget component datasets when applying existing BCC methods at the monthly scale, such as negative P, ET, and Q. These unrealistic negative values arise when an excessive share of the $\Delta$Res is

redistributed to specific components. For instance, if the BCC method overestimates the error in a specific component, it may assign an excessively large portion of ΔRes to that component. When the magnitude of the correction exceeds the component's original value, the result is a negative flux, which is hydrologically incorrect. Beyond introducing negative values, such imbalanced redistribution compromises the integrity of the remaining components. Overcorrecting one variable necessarily reduces the share of ΔRes available for others, potentially degrading their accuracy. Our previous work demonstrated that enforcing water budget closure can, to some extent, reduce the accuracy of individual components and tends to introduce an ET regulation factor to mitigate accuracy loss in ET caused by existing BCC methods (Luo et al., 2023). A more hydrologically sound approach may involve partial closure, whereby only the portion of ΔRes attributable to quantified uncertainties is redistributed, while the residual linked to unmeasured processes is retained.".

(3) We have added further hydrological explanation to justify the design and effectiveness of our proposed IWE-Res method:

"In this section, we propose the IWE-Res method to identify the optimal balance for redistributing ΔRes, minimizing the sum of the introduced error to budget components and the remaining ΔRes error while reducing the negative values introduced by closing the water budget. Unlike existing BCC methods that fully redistribute the ΔRes term in a single step, the IWE-Res method adopts a gradual, iterative redistribution strategy that allows for more consistent correction. Specifically, the method incrementally allocates fractions of ΔRes to P, ET, Q and TWSC, based on fixed percentage steps and guided by existing BCC weighting schemes. At each iteration, the redistribution process seeks to minimize the combined error—defined as the sum of the induced changes in the water budget components and the remaining unexplained ΔRes. This dual-objective criterion ensures that the method balances error reduction while maintaining hydrological plausibility. Importantly, the approach includes a mechanism to avoid introducing implausible negative values. If, during any iteration, the corrected value of a component becomes negative—violating hydrological constraints such as non-negative precipitation or runoff—further redistribution to that component is halted. Subsequent iterations reallocate the remaining ΔRes among the unaffected components. From a hydrological perspective, this strategy acknowledges that not all of the residual can be attributed to known components. Some portion of ΔRes may originate from unmeasured or poorly constrained processes. By partially closing the water budget in a controlled and iterative manner, the IWE-Res method reduces the risk of overcorrecting well-characterized components while better preserving the consistency of the entire budget.".

(4) A comparison with the annual-scale results from the physically based Budyko model indicates that our proposed method produces similar outcomes. Both approaches suggest that the water balance is primarily influenced by P and potential ET (Sankarasubramanian & Vogel, 2002; Zhang et al., 2008; Koster & Suarez, 1999). These influences, however, differ across climatic regions. For instance, in tropical and arid zones, P tends to be the dominant factor (Du et al., 2024; Wu et al., 2018; Liu et al., 2017; Guo et al., 2022), whereas in cold regions, the Budyko model shows relatively low accuracy in estimating ET at the annual scale (Lute et al., 2014; Gao et al., 2010; Potter et al., 2005). This comparison further supports the reliability of our proposed method. The following sentence has been added to the revised manuscript accordingly:

"Previous studies based on the Budyko framework (ignoring TWSC) at the annual scale have shown that water balance is primarily governed by P and potential ET (Sankarasubramanian & Vogel,

2002; Zhang et al., 2008; Koster & Suarez, 1999). However, these influences vary across climatic regions. For example, in tropical and arid regions, P tends to be the dominant controlling factor (Du et al., 2024; Wu et al., 2018; Liu et al., 2017; Guo et al., 2022). In cold regions, the Budyko model exhibits relatively limited accuracy in estimating ET at the annual scale (Lute et al., 2014; Gao et al., 2010; Potter et al., 2005). These previous findings at the annual scale provide indirect support for our results derived at the monthly scale.".

It is not immediately evident to this reviewer that negative values are a problem, especially for the soil storage term (TWSC). In fact one may expect this term to be symmetric around zero, and maybe the same holds for the errors?

**Response:** Thank you for your careful review. We apologize for not clearly explaining in the original manuscript the meaning of negative values for P, ET, and Q, as well as the treatment of "negative" values in TWSC, which may have led to misunderstanding.

In our manuscript, the term "negative values" refers to instances where the values of P, ET, and Q, which were originally positive, become negative after applying a BCC method. For TWSC, we refer to a change in sign—either from positive to negative or vice versa—after correction. These phenomena are caused by the application of BCC methods and intended to highlight a key limitation of existing BCC methods: when the full $\Delta Res$ is redistributed to budget components, it may introduce more uncertainty. Our modifications are as follows:

(1) To avoid any ambiguity, we have revised the manuscript to clarify the meaning of the "negative values" referred to in this study, as follows:

"A clear manifestation of this limitation is the occurrence of negative values in corrected budget component datasets when applying existing BCC methods at the monthly scale, such as negative P, ET, and Q. These unrealistic negative values arise when an excessive share of the $\Delta Res$ is redistributed to specific components. For instance, if the BCC method overestimates the error in a specific component, it may assign an excessively large portion of $\Delta Res$ to that component. When the magnitude of the correction exceeds the component's original value, the result is a negative flux, which is hydrologically incorrect. Beyond introducing negative values, such imbalanced redistribution compromises the integrity of the remaining components. Overcorrecting one variable necessarily reduces the share of $\Delta Res$ available for others, potentially degrading their accuracy. Our previous work demonstrated that enforcing water budget closure can, to some extent, reduce the accuracy of individual components and tends to introduce an ET regulation factor to mitigate accuracy loss in ET caused by existing BCC methods (Luo et al., 2023). A more hydrologically sound approach may involve partial closure, whereby only the portion of $\Delta Res$ attributable to quantified uncertainties is redistributed, while the residual linked to unmeasured processes is retained.".

(2) Regarding TWSC, we fully agree with you that this variable can naturally take both positive and negative values. Indeed, many studies have shown that at annual scales, and with sufficiently long time series, the long-term average of TWSC can be considered negligible. We apologize for not clearly describing how our proposed method handles TWSC during the correction process. In our proposed method, we assume that TWSC observed by the GRACE satellite is reliable. Therefore, if the sign of TWSC changes after correction by the BCC method (e.g., from positive to negative or vice versa), we consider the correction to TWSC to be physically unreasonable, and we suspend further redistribution of $\Delta Res$ to TWSC in subsequent iterations. The following sentence has been

added to the revised manuscript:

"During the iterative correction process, if any of the water budget components (P, ET, and Q) becomes negative, the redistribution of water imbalance error to that component is immediately suspended. In subsequent iterations, redistribution is recalculated to ensure that only components with physically meaningful positive values receive the imbalance correction. For example, if ET becomes negative in a given iteration, the imbalance is subsequently redistributed to P, Q, and TWSC only, in accordance with Equation 33. For TWSC, if a sign reversal occurs during iteration (i.e., from positive to negative or vice versa), the redistribution of the water imbalance error to TWSC is suspended in the following iteration.".

One would assume, based on considerations of the various terms of the water balance that a comparison between the negative (or positive) residuals over time will help identifying which term may be primarily responsible: the soil term can dominate in the short term (e.g. days), but will be small for annual comparisons and may become negligible at decadal scale (except for the long-term desiccation discourse). Spatial patterns are also expected, as frontal rainfall patterns are much easier to represent correctly than thunderstorms (much of tropics and arid zone rainfall) -- indeed your later results (Fig. 6) seem to match this expectation.

**Response:** Thank you very much for your thoughtful and insightful review. We fully agree with your perspective that analyzing the sign and magnitude of residuals over time can provide valuable clues in identifying the dominant contributors to water imbalance errors at different temporal scales. For instance, at daily scales, soil moisture variations may dominate, while at annual or longer time scales, their influence becomes negligible—except in regions experiencing long-term drying trends. Likewise, spatial patterns of residuals reflect regional climatic features, with frontal precipitation systems generally captured more accurately than convective storms in tropical and arid zones.

However, for the purpose of correcting water budget closure at the monthly scale—which is the focus of this study—this method of using sign analysis of residuals over time is less effective in accurately identifying the dominant error source and quantifying the uncertainty of each component. This is due to two key reasons: 1) TWSC cannot be neglected at the monthly scale, meaning the residuals are not solely attributable to errors in P, ET, or Q; 2) Redistribution of residual errors in BCC methods requires quantitative estimation of monthly uncertainties for each component. Unfortunately, time-series-based sign analysis is generally insufficient for this purpose, especially when precise, month-by-month uncertainty quantification is needed. Nonetheless, we acknowledge that residual pattern analysis over time remains a valuable tool for validating the plausibility of error attribution made at the monthly scale. As such, we view it as a complementary approach for guiding and validating the redistribution weights used in monthly BCC corrections.

Currently, most studies on water budget closure correction focus on the monthly scale, for the following reasons: 1) At daily scales, the uncertainty in remote sensing estimates of water cycle variables is even higher than at the monthly scale, and there is a lack of reliable daily-scale data on water storage change (e.g., GRACE data are available only at the monthly scale), which makes daily water balance studies highly uncertain; 2) At annual scales, the long-term average of TWSC is often negligible, and with relatively accurate observations of Q, the sign of the residual can often indicate the balance between P and ET, enabling identification of the primary sources of error. However, if monthly water cycle variables can be corrected to achieve consistent closure, the resulting annual water balance would also be consistent, with the added benefit of preserving the seasonal variability

that is masked in annual-scale analyses. For this reason, the monthly scale remains the primary focus of BCC-related research.

Importantly, we also recognize the potential value of using annual-scale residual analysis as a constraint for monthly correction. For example, within a given basin, the dominant error sources identified at the annual scale should, in principle, align with the redistribution priorities at the monthly scale. To incorporate your suggestions, we have made the following revisions in the manuscript:

(1) Clarified the temporal focus of this study in the introduction: "In this study, we quantify the uncertainties introduced by four existing BCC methods (CKF, MCL, MSD, and PR) at the monthly scale across 84 basins spanning diverse climate zones.".

(2) Expanded the discussion of how spatiotemporal residual patterns can inform future improvements in BCC methods, especially regarding the potential of using annual-scale insights to guide monthly corrections: "Overall, optimizing the redistribution ratio of water imbalance errors is critical for improving the accuracy of corrected budget components. However, the sensitivity of these components to error redistribution varies, and both over- and under-correction can propagate new imbalances across the remaining terms, ultimately misrepresenting the underlying hydrological processes. While existing BCC methods estimate redistribution weights based on the relative uncertainty of each component, future research should examine the physical rationale behind these redistributions. The spatiotemporal variability of residual errors offers valuable insight into their dominant sources, which can serve as an independent reference to validate the influence weights computed by BCC methods. For instance, as shown in previous studies, the contribution of TWSC to residual errors diminishes at annual and especially decadal timescales, where precipitation (P) and evapotranspiration (ET) uncertainties become more dominant. Spatial patterns of residuals also reflect the nature of regional precipitation regimes. In the regions dominated by frontal systems, such as temperate zones, remotely sensed precipitation products tend to capture rainfall events more accurately, leading to smaller residuals. In contrast, in areas characterized by convective rainfall— such as the tropics and arid zones—larger residuals are observed, likely due to the higher uncertainty in capturing short-lived and spatially localized storm events.".

Maybe further reference can be made to the 'Budyko' literature that looks at an annual balance, while your current analysis takes a monthly perspective.

**Response:** Thank you for your constructive suggestions. We have reviewed and summarized relevant literature that applied the Budyko framework at the annual timescale. Many studies have shown that water balance at this scale is primarily influenced by P and potential ET (Sankarasubramanian & Vogel, 2002; Zhang et al., 2008; Koster & Suarez, 1999). These influences, however, differ across climatic regions. For instance, in tropical and arid zones, P tends to be the dominant factor (Du et al., 2024; Wu et al., 2018; Liu et al., 2017; Guo et al., 2022), whereas in cold regions, the Budyko model shows relatively low accuracy in estimating ET at the annual scale (Lute et al., 2014; Gao et al., 2010; Potter et al., 2005). These previous findings indirectly support our results at the monthly scale. The following sentence has been added to the revised manuscript accordingly:

"Previous studies based on the Budyko framework (ignoring TWSC) at the annual scale have shown that water balance is primarily governed by P and potential ET (Sankarasubramanian & Vogel, 2002; Zhang et al., 2008; Koster & Suarez, 1999). However, these influences vary across climatic

regions. For example, in tropical and arid regions, P tends to be the dominant controlling factor (Du et al., 2024; Wu et al., 2018; Liu et al., 2017; Guo et al., 2022). In cold regions, the Budyko model exhibits relatively limited accuracy in estimating ET at the annual scale (Lute et al., 2014; Gao et al., 2010; Potter et al., 2005). These previous findings at the annual scale provide indirect support for our results derived at the monthly scale.".

The abstract could become more attractive to readers if the time unit (monthly balance calculations) is made explicit, as results for daily or annual balance calculations will likely be different.

**Response:** Thank you very much for your constructive suggestion. We have revised the abstract to clarify that this study calculates water balance at the monthly scale: "In this study, we quantify the uncertainties introduced by four existing BCC methods (CKF, MCL, MSD, and PR) at the monthly scale across 84 basins spanning diverse climate zones.".

Details:

The Highlights should be understandable for a non-technical expert -- at the moment they are too full of jargon to attract readers.

**Response:** Thank you very much for your constructive suggestion. We have revised the Highlights by removing technical jargon and replacing them with plain language descriptions. The updated Highlights clearly convey the key findings and significance of the work in a more understandable and engaging manner. Please see our revised Highlights below.

- Existing correction methods may introduce large errors, and more seriously cause unrealistic negative values in P, ET and Q in up to 10% of cases.
- A novel IWE-Res method is proposed to improve the accuracy and consistency of corrected satellite-based water budget component data.
- In most river basins (except cold regions), the best correction is achieved by adjusting 40% to 90% of the total water imbalance error.

Line 57 Indeed a closed budget gives some confidence in the underlying estimates, but not if the closure is obtained by 'fudging' the data, without 'understanding'. So I disagree that 'closing the budget' helps with 'understanding'.

**Response:** Thank you very much for your careful review. We sincerely apologize for the inappropriate expression in the original sentence. Although a large number of datasets for individual budget components have been produced, discrepancies such as measurement errors, systematic biases, and unmeasured components prevent the closure of the water budget among these datasets.

In reality, the components of the water budget are interconnected; together, they regulate the exchange of water among the atmosphere, land, and oceans. Many studies have attempted to reduce the imbalance in the water budget among existing datasets by estimating and correcting errors in the individual components, thereby improving the overall consistency.

We apologize for our unclear expression in our original sentence. On the one hand, we changed the word "understanding" to "confidently applying budget components in hydrological studies"; On the other hand, we have expanded this sentence to clearly explain the concept of the terrestrial water balance, the main components it includes, and their interactions. We then emphasize the importance of improving the consistency among these datasets in order to accurately understand hydrological

processes, due to the intrinsic interconnections among water budget components in real-world hydrological systems, as follows: "The terrestrial water balance represents a fundamental physical framework that describes the distribution and movement of water across the Earth's land surface (Lehmann et al., 2022). It is governed by four interconnected components—precipitation (P), evapotranspiration (ET), streamflow (Q), and terrestrial water storage change (TWSC)—that together regulate the exchange of water among the atmosphere, land, and oceans (Abolafia-Rosenzweig et al., 2021; Sahoo et al., 2011; Chen et al., 2020; Wang et al., 2015). These fluxes and storages are dynamically linked, responding to climatic variability, land surface heterogeneity, and human interventions across a range of spatial and temporal scales. Ensuring a closed water balance is essential for maintaining consistency among these budget components and for confidently applying them in hydrological studies (Li et al., 2024; Mourad et al., 2024).".

Line 66. Before delving into the details it will be good for the reader to be reminded of the physical aspects of uncertainty in the various terms, as these are of different natures:

P precipitation input -- the typically are fairly reliable point data from rainfall gauge data, often with some need t gap fill missing data. The main uncertainty here is in the spatial distribution and representativeness of rainfall gauges, in relation to rainfall types (for frontal rains the spatial uncertainty is low, for local storms it can be high). The distribution of rainfall gauges is often determined in part by accessibility and convenience, and overall uncertainty of daily rainfall may be easily underestimated. More recent satellite based estimates of rainfall appear to perform well for frontal rains, but not in other rainfall types.

ET Evapotranspiration equations have been fairly well calibrated, but there can be uncertainty over the advection term especially in small catchments. For larger areas energy balance equations may be sufficient.

Q monitoring of outflow can have low uncertainty if ;rating curves' are frequently calibrated. However, the delineation of the watershed (and area used for the calculations) can be off where groundwater flows don't necessarily follow surface catchment delineations and can be underestimated.

TWSC can become negligible if a multi-year balance is considered (verifying the P and Q estimates) but can dominate the balance at a daily time-scale. A major challenge is the depth over which TWSC is to be assessed, as changes in the topsoil can be more easily assessed than that deeper in the soil.

**Response:** Thank you very much for your thorough review and constructive suggestions, which have been very helpful in improving the quality and readability of our manuscript. According to your suggestion, we have added a paragraph in the revised manuscript to describe the physical aspects of uncertainty in the four water cycle components (P, ET, Q, and TWSC) as follows:

"P is typically derived from point-based rain gauge networks, which are generally reliable but often incomplete, requiring gap-filling (Esquivel-Arriaga et al., 2024; Nassaj et al., 2022; Bai et al., 2021; Lockhoff et al., 2014). The main source of uncertainty lies in the spatial distribution and representativeness of these gauges, particularly in relation to precipitation type (Bai et al., 2019; Trenberth et al., 2014). Spatial uncertainty tends to be low for widespread frontal systems but can be substantial for localized convective storms (Palharini et al., 2020). Gauge placement is often dictated by accessibility and logistical convenience, which may lead to underestimation of the uncertainty in daily precipitation inputs (Wang et al., 2017; Bai et al., 2019; Wu et al., 2018). Satellite-based precipitation estimates have demonstrated good performance in capturing frontal

rainfall, but not in other rainfall types (Masunaga et al., 2019; Petković et al., 2017; Palharini et al., 2020). ET is commonly estimated using empirical or physically based models (Jacobs et al., 1998; McMahon et al., 2016; Allen et al., 1998). Although these models are generally well calibrated, uncertainties persist due to the complex influence of advection and localized meteorological variability, especially in small catchments. At larger spatial scales, energy balance approaches tend to provide sufficiently accurate estimates (Hua et al., 2020; Hao et al., 2018; Ruhoff et al., 2022). Q measurements typically exhibit low uncertainty when rating curves are well established and regularly maintained (Jian et al., 2015; Krabbenhoft et al., 2022). However, uncertainty can still arise from the delineation of watershed boundaries, particularly in regions where groundwater flow does not align with surface catchment divides (Huang et al., 2023; Bouaziz et al., 2018). This mismatch can result in misrepresentation of actual hydrological contributions. TWSC generally has a negligible impact on water balance calculations over multi-year periods, but can significantly affect short-term (e.g., daily) balances (He et al., 2023; Zhang et al., 2016). A key challenge is defining the effective depth over which TWSC should be quantified, as changes in soil moisture near the surface are more easily observed than those occurring at greater depths.".

Line 98 Negative ET is possible under 'dew formation' conditions... (be it in only part of a daily temperature cycle)

**Response:** Thank you for your careful review. We apologize for not clearly explaining in the original manuscript the meaning of negative values for P, ET, and Q, which may have led to misunderstanding. In our manuscript, the term "negative values" refers to instances where the values of P, ET, and Q, which were originally positive, become negative after applying a BCC method. These phenomena are caused by the application of BCC methods and intended to highlight a key limitation of existing BCC methods: when the full ΔRes is redistributed to budget components, it may introduce more uncertainty. Our modifications are as follows:

To avoid any ambiguity, we have revised the manuscript to clarify the meaning of the "negative values" referred to in this study, as follows: "A clear manifestation of this limitation is the occurrence of negative values in corrected budget component datasets when applying existing BCC methods at the monthly scale, such as negative P, ET, and Q. These unrealistic negative values arise when an excessive share of the ΔRes is redistributed to specific components. For instance, if the BCC method overestimates the error in a specific component, it may assign an excessively large portion of ΔRes to that component. When the magnitude of the correction exceeds the component's original value, the result is a negative flux, which is hydrologically unrealistic. Beyond introducing negative values, such imbalanced redistribution compromises the integrity of the remaining components. Overcorrecting one variable necessarily reduces the share of ΔRes available for others, potentially degrading their accuracy. Our previous work demonstrated that enforcing water budget closure can, to some extent, reduce the accuracy of individual components and tends to introduce an ET regulation factor to mitigate accuracy loss in ET caused by existing BCC methods (Luo et al., 2023). A more hydrologically sound approach may involve partial closure, whereby only the portion of ΔRes attributable to quantified uncertainties is redistributed, while the residual linked to unmeasured processes is retained.".

Line 244 There can be 'bias' (systematic error, e.g. if groundwater flows mean that the basin is not closed and part of outflowing Q is missed; the area of the basin can also be incorrect), part

'measurement error'. As you focus on relatively large basins, the bias term may be relatively small, but for smaller watersheds the bias terms cannot be ignored. Standard techniques such as plotting cumulative Q vs cumulative P give indications, especially if nested Q data exist beyond outflow data.

**Response:** Thank you very much for your kind reminder. In the revised manuscript, we have added a description regarding systematic biases, as follows: "However, in practice, this balance is rarely achieved due to various sources of error. These include systematic biases (such as missed portions of outflow resulting from unclosed basin boundaries and inaccuracies in catchment area delineation, particularly in small basins), measurement uncertainties, and the omission of unmeasured components. Consequently, each budget component (P, ET, Q, and TWSC) is subject to an associated error term (denoted as $\varepsilon_P$, $\varepsilon_{ET}$, $\varepsilon_Q$, $\varepsilon_{TWSC}$, respectively), leading to a non-closure of the water budget (i.e., Equation 1 becomes Equation 4) (Aires, 2014; Wong et al., 2021).".

Line 702-714, Figure 7 - would it make sense to compensate S Hemisphere data for a 6 month shift in seasons? Or even more flexibly to use a hydrological year concept with a standardized month for maximum P.

**Response:** Thank you for the insightful suggestion. We agree that seasonal misalignment between the Northern and Southern Hemispheres may obscure underlying hydrological patterns. We have redrawn Figure 7 accordingly. Specifically, based on your recommendation, we adjusted the Southern Hemisphere data by applying a 6-month shift to align its seasonal phases with those of the Northern Hemisphere. This adjustment has been noted in the caption of the revised Figure 7. Finally, we reanalyzed the figure based on the updated version.

"Fig. 7 presents the seasonal cycle of negative values across different climate zones, examining whether these values exhibit significant seasonal patterns. Negative P values predominantly occur in winter and spring, with a higher proportion from January to March in tropical climates compared to arid regions. ET tends to show negative values more frequently in winter and spring, with a lower likelihood in summer and autumn. Except in summer, cold climate zones are most susceptible to negative ET values. Among the four budget components, Q has the lowest occurrence of negative values. Negative TWSC values exhibit no obvious seasonal pattern, with arid regions exhibiting a higher likelihood of negative values throughout the year compared to other climate types. These findings indicate that the occurrence of negative values varies significantly across seasons and climate zones. Future research should account for this seasonal variability to further refine existing BCC methods.".

[Figure]

**Fig. 7** Seasonal cycle of the proportion of negative errors for budget components. Different colors representing various climate types. The Southern Hemisphere data by applying a 6-month shift to align its seasonal phases with those of the Northern Hemisphere.

Reference:

Abatzoglou, J. T., Dobrowski, S. Z., Parks, S. A., & Hegewisch, K. C. (2018). TerraClimate, a high-resolution global dataset of monthly climate and climatic water balance from 1958–2015. Scientific Data, 5(1), 1-12. https://doi.org/10.1038/sdata.2017.191

Aires, F. (2014). Combining datasets of satellite-retrieved products. Part I: Methodology and water budget closure. Journal of Hydrometeorology, 15(4), 1677-1691. https://doi.org/10.1175/JHM-D-13-0148.1

Allen, R. G., Pereira, L. S., Raes, D., & Smith, M. (1998). Crop evapotranspiration-Guidelines for computing crop water requirements-FAO Irrigation and drainage paper 56. Fao, Rome, 300(9), D05109.

Bai, X., Wu, X., & Wang, P. (2019). Blending long-term satellite-based precipitation data with gauge observations for drought monitoring: Considering effects of different gauge densities. Journal of Hydrology, 577, 124007. https://doi.org/10.1016/j.jhydrol.2019.124007

Bai, X., Wang, P., He, Y., Zhang, Z., & Wu, X. (2021). Assessing the accuracy and drought utility of long-term satellite-based precipitation estimation products using the triple collocation approach. Journal of Hydrology, 603, 127098. https://doi.org/10.1016/j.jhydrol.2021.127098

Bouaziz, L., Weerts, A., Schellekens, J., Sprokkereef, E., Stam, J., Savenije, H., & Hrachowitz, M. (2018). Redressing the balance: quantifying net intercatchment groundwater flows. Hydrology and Earth System Sciences, 22(12), 6415-6434. https://doi.org/10.5194/hess-22-6415-2018

Chen, J., Tapley, B., Rodell, M., Seo, K. W., Wilson, C., Scanlon, B. R., & Pokhrel, Y. (2020). Basin-scale river runoff estimation from GRACE gravity satellites, climate models, and in situ observations: A case study in the Amazon basin. Water resources research, 56(10), e2020WR028032. https://doi.org/10.1029/2020WR028032

Du, H., Zeng, S., Liu, X., & Xia, J. (2024). An improved Budyko framework model incorporating water-carbon relationship for estimating evapotranspiration under climate and vegetation changes. Ecological Indicators, 169, 112887. https://doi.org/10.1016/j.ecolind.2024.112887

Esquivel-Arriaga, G., Huber-Sannwald, E., Reyes-Gómez, V. M., Bravo-Peña, L. C., Dávila-Ortiz, R., Martínez-Tagüeña, N., & Velázquez-Zapata, J. A. (2024). Performance Evaluation of Global Precipitation Datasets in Northern Mexico Drylands. Journal of Applied Meteorology and Climatology, 63(12), 1545-1558. https://doi.org/10.1175/JAMC-D-23-0227.1

Gao, H., Tang, Q., Ferguson, C. R., Wood, E. F., & Lettenmaier, D. P. (2010). Estimating the water budget of major US river basins via remote sensing. International Journal of Remote Sensing, 31(14), 3955-3978. https://doi.org/10.1080/01431161.2010.483488

Guo, W., Hong, F., Yang, H., Huang, L., Ma, Y., Zhou, H., & Wang, H. (2022). Quantitative evaluation of runoff variation and its driving forces based on multi-scale separation framework. Journal of Hydrology: Regional Studies, 43, 101183. https://doi.org/10.1016/j.ejrh.2022.101183

Hao, X., Zhang, S., Li, W., Duan, W., Fang, G., Zhang, Y., & Guo, B. (2018). The uncertainty of Penman-Monteith method and the energy balance closure problem. Journal of Geophysical Research: Atmospheres, 123(14), 7433-7443. https://doi.org/10.1029/2018JD028371

He, Q., Fok, H. S., Ferreira, V., Tenzer, R., Ma, Z., & Zhou, H. (2023). Three-dimensional Budyko framework incorporating terrestrial water storage: Unraveling water-energy dynamics, vegetation, and ocean-atmosphere interactions. Science of the Total Environment, 904, 166380. https://doi.org/10.1016/j.scitotenv.2023.166380

Hua, D., Hao, X., Zhang, Y., & Qin, J. (2020). Uncertainty assessment of potential evapotranspiration in arid areas, as estimated by the Penman-Monteith method. Journal of Arid Land, 12, 166-180. https://doi.org/10.1007/s40333-020-0093-7

Huang, P., Wang, G., Guo, L., Mello, C. R., Li, K., Ma, J., & Sun, S. (2023). Most global gauging stations present biased estimations of total catchment discharge. Geophysical Research Letters, 50(15), e2023GL104253. https://doi.org/10.1029/2023GL104253

Jacobs, A. F. G., & De Bruin, H. A. R. (1998). Makkink's equation for evapotranspiration applied to unstressed maize. Hydrological processes, 12(7), 1063-1066. https://doi.org/10.1002/(SICI)1099-1085(19980615)12:7<1063::AID-HYP640>3.0.CO;2-2

Jian, J., Ryu, D., Costelloe, J. F., & Su, C. H. (2015). Towards reliable hydrological model calibrations with river level measurements. In 21st International Congress on Modelling and Simulation, Modelling and Simulation Society of Australia and New Zealand.

Koster, R. D., & Suarez, M. J. (1999). A simple framework for examining the interannual variability of land surface moisture fluxes. Journal of Climate, 12(7), 1911-1917. https://doi.org/10.1175/1520-0442(1999)012<1911:ASFFET>2.0.CO;2

Krabbenhoft, C. A., Allen, G. H., Lin, P., Godsey, S. E., Allen, D. C., Burrows, R. M., ... & Olden, J. D. (2022). Assessing placement bias of the global river gauge network. Nature Sustainability, 5(7), 586-592. https://doi.org/10.1038/s41893-022-00873-0

Lehmann, F., Vishwakarma, B. D., & Bamber, J. (2022). How well are we able to close the water budget at the global scale?. Hydrology and Earth System Sciences, 26(1), 35-54. https://doi.org/10.5194/hess-26-35-2022

Li, L., Dai, Y., Wei, Z., Wei, S., Zhang, Y., Wei, N., & Li, Q. (2024). Enforcing Water Balance in Multitask Deep Learning Models for Hydrological Forecasting. Journal of Hydrometeorology,

25(1), 89-103. https://doi.org/10.1175/JHM-D-23-0073.1

Liu, J., Zhang, Q., Singh, V. P., & Shi, P. (2017). Contribution of multiple climatic variables and human activities to streamflow changes across China. Journal of Hydrology, 545, 145-162.

Lockhoff, M., Zolina, O., Simmer, C., & Schulz, J. (2014). Evaluation of satellite-retrieved extreme precipitation over Europe using gauge observations. Journal of climate, 27(2), 607-623. https://doi.org/10.1175/JCLI-D-13-00194.1

Luo, Z., Gao, Z., Wang, L., Wang, S., & Wang, L. (2023). A method for balancing the terrestrial water budget and improving the estimation of individual budget components. Agricultural and Forest Meteorology, 341, 109667. https://doi.org/10.1016/j.agrformet.2023.109667

Luo, Z., Li, H., Zhang, S., Wang, L., Wang, S., & Wang, L. (2023). A novel two-step method for enforcing water budget closure and an intercomparison of budget closure correction methods based on satellite hydrological products. Water Resources Research, 59(3), e2022WR032176. https://doi.org/10.1029/2022WR032176

Lute, A. C., & Abatzoglou, J. T. (2014). Role of extreme snowfall events in interannual variability of snowfall accumulation in the western United States. Water Resources Research, 50(4), 2874-2888. https://doi.org/10.1002/2013WR014465

Masunaga, H., Schröder, M., Furuzawa, F. A., Kummerow, C., Rustemeier, E., & Schneider, U. (2019). Inter-product biases in global precipitation extremes. Environmental Research Letters, 14(12), 125016. https://doi.org/ 10.1088/1748-9326/ab5da9

Mourad, R., Schoups, G., Bastiaanssen, W., & Kumar, D.N. (2024). Expert-based prior uncertainty analysis of gridded water balance components: Application to the irrigated Hindon River Basin, India. Journal of Hydrology: Regional Studies, 55, 2214-5818. https://doi.org/10.1016/j.ejrh.2024.101935.

McMahon, T. A., Finlayson, B. L., & Peel, M. C. (2016). Historical developments of models for estimating evaporation using standard meteorological data. Wiley Interdisciplinary Reviews: Water, 3(6), 788-818. https://doi.org/10.1002/wat2.1172

Munier, S., & Aires, F. (2018). A new global method of satellite dataset merging and quality characterization constrained by the terrestrial water budget. Remote Sensing of Environment, 205, 119-130. https://doi.org/10.1016/j.rse.2017.11.008

Nassaj, B. N., Zohrabi, N., Shahbazi, A. N., & Fathian, H. (2022). Evaluating the performance of eight global gridded precipitation datasets across Iran. Dynamics of Atmospheres and Oceans, 98, 101297. https://doi.org/10.1016/j.dynatmoce.2022.101297

Palharini, R. S. A., Vila, D. A., Rodrigues, D. T., Quispe, D. P., Palharini, R. C., de Siqueira, R. A., & de Sousa Afonso, J. M. (2020). Assessment of the extreme precipitation by satellite estimates over South America. Remote Sensing, 12(13), 2085. https://doi.org/10.3390/rs12132085

Petković, V., & Kummerow, C. D. (2017). Understanding the sources of satellite passive microwave rainfall retrieval systematic errors over land. Journal of Applied Meteorology and Climatology, 56(3), 597-614. https://doi.org/10.1175/JAMC-D-16-0174.1

Pan, M., Sahoo, A. K., Troy, T. J., Vinukollu, R. K., Sheffield, J., & Wood, E. F. (2012). Multisource estimation of long-term terrestrial water budget for major global river basins. Journal of Climate, 25(9), 3191-3206. https://doi.org/10.1175/JCLI-D-11-00300.1

Potter, N. J., Zhang, L., Milly, P. C. D., McMahon, T. A., & Jakeman, A. J. (2005). Effects of rainfall seasonality and soil moisture capacity on mean annual water balance for Australian catchments. Water Resources Research, 41(6). https://doi.org/10.1029/2004WR003697

Ruhoff, A., de Andrade, B. C., Laipelt, L., Fleischmann, A. S., Siqueira, V. A., Moreira, A. A., ... & Biggs, T. (2022). Global evapotranspiration datasets assessment using water balance in South America. Remote Sensing, 14(11), 2526. https://doi.org/10.3390/rs14112526

Sahoo, A. K., Pan, M., Troy, T. J., Vinukollu, R. K., Sheffield, J., & Wood, E. F. (2011). Reconciling the global terrestrial water budget using satellite remote sensing. Remote Sensing of Environment, 115(8), 1850-1865. https://doi.org/10.1016/j.rse.2011.03.009

Sankarasubramanian, A., & Vogel, R. M. (2002). Annual hydroclimatology of the United States. Water Resources Research, 38(6), 19-1. https://doi.org/10.1029/2001WR000619

Trenberth, K. E., Dai, A., Van Der Schrier, G., Jones, P. D., Barichivich, J., Briffa, K. R., & Sheffield, J. (2014). Global warming and changes in drought. Nature Climate Change, 4(1), 17-22. https://doi.org/10.1038/nclimate2067

Wang, S., Huang, J., Yang, D., Pavlic, G., & Li, J. (2015). Long-term water budget imbalances and error sources for cold region drainage basins. Hydrological Processes, 29(9), 2125-2136. https://doi.org/10.1002/hyp.10343

Wang, Z., Zhong, R., Lai, C., & Chen, J. (2017). Evaluation of the GPM IMERG satellite-based precipitation products and the hydrological utility. Atmospheric Research, 196, 151-163. https://doi.org/10.1016/j.atmosres.2017.06.020

Wong, J. S., Zhang, X., Gharari, S., Shrestha, R. R., Wheater, H. S., & Famiglietti, J. S. (2021). Assessing water balance closure using multiple data assimilation–and remote sensing–based datasets for canada. Journal of Hydrometeorology, 22(6), 1569-1589. https://doi.org/10.1175/JHM-D-20-0131.1

Wu, Z., Zhang, Y., Sun, Z., Lin, Q., & He, H. (2018). Improvement of a combination of TMPA (or IMERG) and ground-based precipitation and application to a typical region of the East China Plain. Science of the Total Environment, 640, 1165-1175. https://doi.org/10.1016/j.scitotenv.2018.05.272

Wu, C., Yeh, P. J. F., Hu, B. X., & Huang, G. (2018). Controlling factors of errors in the predicted annual and monthly evaporation from the Budyko framework. Advances in water resources, 121, 432-445. https://doi.org/10.1016/j.advwatres.2018.09.013

Zhang, D., Liu, X., Zhang, Q., Liang, K., & Liu, C. (2016). Investigation of factors affecting intra-annual variability of evapotranspiration and streamflow under different climate conditions. Journal of Hydrology, 543, 759-769. https://doi.org/10.1016/j.jhydrol.2016.10.047

Zhang, L., Potter, N., Hickel, K., Zhang, Y., & Shao, Q. (2008). Water balance modeling over variable time scales based on the Budyko framework–Model development and testing. Journal of Hydrology, 360(1-4), 117-131. https://doi.org/10.1016/j.jhydrol.2008.07.021

Zhang, Y., Pan, M., & Wood, E. F. (2016). On creating global gridded terrestrial water budget estimates from satellite remote sensing. Remote sensing and water resources, 59-78. https://doi.org/10.1007/978-3-319-32449-4_4

Zhou, L., Cao, Y., Shi, C., Liang, H., & Fan, L. (2024). Quantifying the Atmospheric Water Balance Closure over Mainland China Using Ground-Based, Satellite, and Reanalysis Datasets. Atmosphere, 15(4), 497. https://doi.org/10.3390/atmos15040497

---

## Author Comment (AC2)

Response to Review
Reviewer #CC1:

   With the rapid development of satellite remote sensing technology, a large number of datasets related to water cycle variables have been produced, providing important opportunities for more accurately revealing hydrological variation processes within watersheds. However, many datasets on precipitation, evapotranspiration, runoff, and water storage change are observed or modeled independently, and contain certain uncertainties. This leads to poor physical consistency among the datasets, often manifested as non-closure of the water balance. It is therefore crucial to obtain consistent datasets.

   This paper addresses the limitations of existing water balance closure correction methods that fully allocate water imbalance residuals, which may lead to negative corrected values. The authors propose a method based on partially closed water balance correction, which effectively resolves the issues of accuracy loss and negative values inherent in current methods. This approach demonstrates important innovation and research value. To help the authors further improve the manuscript, I have the following suggestions:

**Response:** Thank you for your careful review. These comments are very helpful for us to improve the manuscript. We really appreciate your time and efforts. The point-to-point responses were given after individual comments.

- It is recommended that the authors provide a more detailed explanation in the introduction regarding the sources of water balance residuals (including errors, unmeasured components, and biases). This would better clarify the irrationality of fully allocating the residuals based solely on observational errors in existing closure methods, as well as the underlying reason why this can easily lead to negative values.

**Response:** Thank you for your constructive suggestion. We apologize for not clearly explaining the sources of water imbalance errors in our manuscript. These errors primarily stem from the estimation errors of budget components (errors in remote sensing and reanalysis products of budget components), systematic biases, and the omission of unmeasured components. However, existing BCC methods typically assume that all of $\Delta$Res arises from estimation errors and then redistribute it according to estimation errors. In the revised manuscript, we have provided a more detailed description of the principles behind existing BCC methods, the main uncertainty sources causing water imbalance error, and the limitations of existing BCC methods that fully redistribute the water imbalance error. The following sentence has been added to the revised manuscript:

   "Existing BCC methods redistribute the entire $\Delta$Res error among water budget components to enforce strict water budget closure. This redistribution is typically guided by the relative uncertainties of the individual components, based on the assumption that the entire residual error originates from observational or modeling errors in these datasets. However, this assumption overlooks the fact that $\Delta$Res is not solely the result of measurement or estimation errors in P, ET, Q, or TWSC. Rather, it is a composite residual that also reflects contributions from systematic biases and the omission of unmeasured components. These include deep groundwater exchanges that may cross basin boundaries, snow and glacier storage changes (particularly in high-altitude or high-latitude regions), and anthropogenic influences such as irrigation withdrawals, reservoir operations, and inter-basin water transfers. Because existing BCC methods do not explicitly account for these

additional sources of imbalance, forcing strict closure by allocating the entire ΔRes to the measured components can introduce unrealistic uncertainties. As a result, the application of existing BCC methods—despite their goal of improving internal consistency — often leads to limited improvements, or, in some cases, even a decline in the accuracy of the corrected hydrological datasets.".

- In the methods section, the authors propose a stepwise iterative approach to find an optimal balance point for allocating residuals, which is a sound strategy. The condition for terminating the iteration is the emergence of negative values in the water cycle variables. While it is easy to understand how precipitation, evapotranspiration, and runoff can become negative, water storage change inherently includes both positive and negative values. Therefore, the authors are advised to clearly explain how the termination condition is defined for water storage change.

**Response:** Thank you for your careful review. We apologize for not clearly explaining the termination condition for the iterative adjustment of terrestrial water storage change (TWSC). We consider it inappropriate to continue redistributing the water imbalance error to TWSC if its sign changes after applying the existing BCC method—specifically, if TWSC shifts from a positive to a negative value, or vice versa. Such a change indicates that the correction applied to TWSC may be unreasonable. This is because GRACE satellite observations of TWSC are generally regarded as reliable, and the primary purpose of BCC methods is to improve the internal consistency among precipitation, evapotranspiration, runoff, and TWSC. A reversal in the sign of TWSC during the correction process suggests a potential overcorrection, and further adjustments should therefore be terminated. We have added the following sentence to the revised manuscript:

"During the iterative correction process, if any of the water budget components (P, ET, and Q) becomes negative, the redistribution of water imbalance error to that component is immediately suspended. In subsequent iterations, redistribution is recalculated to ensure that only components with physically meaningful positive values receive the imbalance correction. For example, if ET becomes negative in a given iteration, the imbalance is subsequently redistributed to P, Q, and TWSC only, in accordance with Equation 33. For TWSC, if a sign reversal occurs during iteration (i.e., from positive to negative or vice versa), the redistribution of the water imbalance error to TWSC is suspended in the following iteration.".

- The paper demonstrates that applying existing water balance closure correction methods may reduce the accuracy of water cycle variables. Although this is an important finding, it is recommended that the authors add a discussion on the underlying reasons for this issue.

**Response:** Thanks for your constructive suggestion. We believe that the reduced accuracy of the corrected datasets resulting from existing water budget closure correction (BCC) methods may be due to the following reasons: 1) Most existing BCC methods estimate errors in budget components without incorporating independent observational data. Inaccurate error estimates for a single budget component can propagate through the redistribution process, biasing the redistribution of the water imbalance error to other budget components and ultimately reducing the accuracy of all budget components (Abolafia-Rosenzweig et al., 2021). Incorporating high-quality observational data into the error estimation process is therefore essential to improve the robustness of BCC methods; 2) Existing BCC methods are limited by the assumption that the entire water imbalance error can be

fully attributed to estimation errors in budget components. These methods enforce water budget closure by completely redistributing the water imbalance error among the budget components. However, this residual may also arise from systematic biases and unmeasured processes, rather than solely from component-level estimation errors. In this study, we propose an iterative optimization approach that seeks a more balanced redistribution of water imbalance error, aiming to minimize both the errors introduced into individual budget components and the remaining water imbalance error. This method significantly improves the accuracy of the corrected datasets. Future research could further enhance this approach by integrating it with physically based hydrological or land surface models, offering a promising pathway toward greater physical realism and internal consistency in corrected water budget datasets; 3) Observational datasets themselves often do not strictly satisfy water budget closure due to inherent measurement limitations and sampling errors. This introduces uncertainty when using these datasets to validate the accuracy of BCC-corrected results. For example, even if the corrected datasets more closely approximate the true values of individual components, the absence of ground-truth observations presents a fundamental challenge for objectively assessing the validity of these corrections. Future studies should prioritize the development of more objective and physically grounded evaluation metrics to better assess the performance of BCC-corrected datasets. We have discussed these issues and potential solutions in the revised manuscript as follows:

"Several factors may contribute to this reduction in accuracy. First, most existing BCC methods estimate errors in budget components without incorporating independent observational data. These methods then redistribute water imbalance errors based on these internally estimated uncertainties (Section 3.2). However, the absence of observational constraints undermines the reliability of the estimated component errors, which in turn leads to a suboptimal and potentially biased allocation of the imbalance. As previously noted, inaccurate error estimates for a single variable can propagate through the redistribution process, biasing the residual redistribution to the remaining budget components and ultimately lowering the accuracy of all water budget components (Abolafia-Rosenzweig et al., 2021). Incorporating high-quality observational data into the error estimation process is therefore essential to improve the robustness of BCC methods; Second, existing BCC methods are limited by their assumption that the entire water imbalance error can be fully attributed to errors in the measured budget components. These methods enforce water budget closure by completely redistributing the water imbalance error among the budget components, yet this residual may also stem from systematic biases and unmeasured processes—not just estimation errors of measured budget components. In this study, we propose an iterative optimization approach that seeks a balanced redistribution of the ΔRes, aiming to minimize both the errors introduced to individual budget components and the remaining ΔRes. This method significantly improves the accuracy of the corrected datasets. Future research may further enhance this framework by integrating it with physically based hydrological or land surface models, which could provide a promising pathway toward more physically consistent and realistic water budget estimates; Third, the observational datasets themselves often fail to strictly satisfy water budget closure due to measurement limitations and sampling errors. This introduces uncertainty when using these datasets to validate the accuracy of BCC-corrected estimates. For instance, even if the corrected datasets more closely approximate the true values of budget components, the lack of ground-truth observations presents a fundamental challenge for objectively evaluating the effectiveness of these corrections. Future work should prioritize the development of more objective and physically

grounded evaluation metrics to assess the accuracy of BCC-corrected datasets. Although this challenge lies beyond the scope of the present study, addressing it will be critical for advancing the reliability of water budget assessments.".

- Some minor formatting issues should be carefully checked. For example, multiple terms and abbreviations are used throughout the paper. It is suggested that the full name along with the abbreviation be given at first mention, with the abbreviation used thereafter.

**Response:** Thank you very much for your careful review. We have thoroughly revised the manuscript to ensure that all abbreviations are spelled out in full when first introduced, with the corresponding abbreviations provided in parentheses. Thereafter, only the abbreviations are used consistently throughout the text.

- The axis labels in Figures 12–15 should be formatted consistently with the other figures. They should not be bolded.

**Response:** Thank you very much for your careful review. We have changed the axis labels in Figures 12–15 from bold to regular font. We have also checked all the figures to ensure consistency.

Reference:

Abolafia-Rosenzweig, R., Pan, M., Zeng, J., & Livneh, B. (2021). Remotely sensed ensembles of the terrestrial water budget over major global river basins: An assessment of three closure techniques. Remote Sensing of Environment, 252, 112191. https://doi.org/10.1016/j.rse.2020.112191

---

## Author Comment (AC3)

Response to Review
Reviewer #2:

Greetings. The manuscript entitled "A novel method for 1 correcting water budget components and reducing their uncertainties by optimally distributing the imbalance residual without full closure" deals with the closure of water budget problems, and specifically with uncertainties therein. The structure and goals are clear, and the results offering is well-suited. This paper can for sure be published after some adjustments, listed below. I think that these itemized improvements would make the work more scientifically sound and robust. These considerations come from my expertise as a hydrogeologist, so they will pertain to this sphere of competency. Best regards.

**Response:** Thank you very much for your careful review. These comments are very helpful for us to improve the manuscript. We really appreciate your time and efforts.

The point-to-point responses were given after comments. All the changes were highlighted in bright blue for easy review. We hope that the revision meets the requirement for publication.

- In equation 2, jkl should be written as subscripts, as well as 123, etc in eq. 3.

**Response:** Thanks for your careful review. We apologize for our carelessness. In the revised manuscript, "jkl" in Equation 2 and "123" in Equation 3 were formatted as subscripts, respectively, as shown below:

"

$$C_{jkl} = \begin{bmatrix} P_j & ET_k & TWSC_l & Q \end{bmatrix} \tag{2}$$

where $j$, $k$, and $l$ represent the indices of the datasets corresponding to each budget component. Table 1 provides basic information on the datasets used in this study, along with their corresponding indices. Equation 3 represents a matrix composed of the elements defined in Equation 2.

$$C = \begin{bmatrix} C_{111} & C_{112} & C_{113} & C_{121} & C_{122} & C_{123} & C_{131} & C_{132} & C_{133} \\ C_{211} & C_{212} & C_{213} & C_{221} & C_{222} & C_{223} & C_{231} & C_{232} & C_{233} \\ C_{311} & C_{312} & C_{313} & C_{321} & C_{322} & C_{323} & C_{331} & C_{332} & C_{333} \\ C_{411} & C_{412} & C_{413} & C_{421} & C_{422} & C_{423} & C_{431} & C_{432} & C_{433} \end{bmatrix} \tag{3}$$

".

We carefully reviewed all equations and notations throughout the manuscript to ensure that all similar issues were addressed.

- We need a more detailed specification, both in the introduction and in the methodology (e.g., from line 64 on) of the TWSC term. It is not sufficient to describe the ground-underground part of the water cycle. Two major points should be at least touched: (i) a portion of the TWSC term is the water infiltrating to aquifer, but that is returned to major water bodies soon or later (see e.g. Levison et al., 2016); (ii) the major role in the aquifer ability to store or drain the portion of water infiltrating is played by local geology, precisely its spatial distribution and the eventual presence of fractures (again, Levison et al., 2016) or highly permeable conduits (Schiavo, 2023). I think these two major points should be supported leveraging the suggested references.

**Response:** Thank you for your constructive suggestion. We sincerely apologize for the unclear description of the TWSC term in the original manuscript. In this study, TWSC refers to the total terrestrial water storage change, which includes but is not limited to surface water, soil moisture,

groundwater, aquifer infiltration, and ice/snow components. To better represent this, we employed three different GRACE satellite-derived datasets to reduce the uncertainty of TWSC estimates.

We fully agree with you that it is insufficient to describe only surface and subsurface part of the water cycle. We should also clarify that the TWSC term includes water that infiltrates into aquifers. This water has both storage and drainage characteristics and eventually contributes to major water bodies, thereby affecting TWSC. In particular, the ability of aquifers to store or release water is strongly influenced by local geological conditions, such as spatial heterogeneity, the presence of fracture zones, and high-permeability pathways. These key aspects were incorporated into the revised manuscript, and we cited the references you suggested as essential support. Specifically, we made revisions in the following three aspects:

(1) We revised the definition of TWSC in the Introduction to clearly state that it includes surface water, soil moisture, groundwater, water infiltrating into aquifers, and ice/snow (Mehrnegar et al., 2023; Pellet et al., 2020; Wang et al., 2022). We referenced the works of Levison et al. (2016) and Schiavo (2023) to emphasize the drainage characteristics of infiltrated aquifer water (eventually returning to major water bodies) and its influencing factors (e.g., geological conditions). Specifically, we added the following sentence to the revised manuscript:

"where P represents precipitation, ET represents evapotranspiration, Q represents streamflow, and TWSC represents terrestrial water storage change. It is worth noting that TWSC refers to the change in total terrestrial water storage, including but not limited to surface water, soil moisture, groundwater, water infiltrating into aquifers, and ice/snow (Mehrnegar et al., 2023; Pellet et al., 2020; Wang et al., 2022). Infiltrated water into aquifers does not remain permanently stored, but is eventually returned to major water bodies sooner or later (Levison et al., 2016). The ability of aquifers to retain or transmit infiltrated water is strongly influenced by local geological characteristics, particularly the spatial heterogeneity, presence of fractures, or high-permeability pathways (Levison et al., 2016; Schiavo, 2023).".

(2) We added a more detailed description of the GRACE TWSC data in the Data section of the manuscript. By briefly explaining the principle by which GRACE satellites monitor TWSC, we clarified that the GRACE-derived TWSC represents total terrestrial water storage change. The following sentence was added to the revised manuscript:

"The launch of the GRACE and GRACE Follow-On (GRACE-FO) satellite missions has provided new opportunities for more accurate observations of large-scale TWSC. GRACE operated from 2002 to 2017, followed by GRACE-FO starting in 2018 (Boergens et al., 2024). These missions infer terrestrial total TWSC by tracking temporal variations in Earth's gravity field, which are primarily attributed to changes in terrestrial water mass. The GRACE TWSC datasets used in this study are provided by the University of Texas Center for Space Research (CSR), the German Research Centre for Geosciences (GFZ), and NASA's Jet Propulsion Laboratory (JPL), all of which include multiple bias correction procedures to improve data quality (Landerer et al., 2012; Shamsudduha et al., 2017). These bias correction procedures include filtration to suppress correlated noise and striping artifacts (Swenson et al., 2006), replacement of poorly resolved spherical harmonic coefficients (e.g., degree-2 term C20) with satellite laser ranging data (Loomis et al., 2020), and correction for glacial isostatic adjustment (GIA) (Peltier et al., 2012; Mu et al., 2017). Numerous studies have demonstrated the sensitivity and reliability of GRACE satellite data for monitoring TWSC (Swenson and Wahr, 2006; Resende et al., 2019; Majid et al., 2019; Reager et al., 2014).".

(3) In the Methodology section, we explicitly emphasized that the TWSC data used in the BCC method refer to the total terrestrial water storage change observed by the GRACE satellite:

"In the following application of the BCC method, the TWSC data used in this study refer to the basin-scale total terrestrial water storage change observed by GRACE satellite data.".

- Global precipitation models, as well as other kinds of climate products, need to be bias corrected to be employable, even if these issues and the opportunity of such procedures are still subject of scientific debate (e.g., Ehret et al., 2012). Are the employed data raw or bias corrected (if so, how)?

**Response:** Thank you very much for your careful review, which has been highly valuable in improving the quality of our manuscript. To more reliably analyze the uncertainties introduced by existing BCC methods and to verify the robustness of the IWE-Res method proposed in this study, we selected multiple datasets for each budget component, forming multiple data combinations in each basin. Specifically, we used four P datasets (GPCC, GPM IMERG Final Run, MSWEP, and PERSIANN-CDR), three ET datasets (GLDAS, GLEAM, and TerraClimate), one observed Q dataset, and three GRACE-based TWSC datasets.

We apologize for not clearly stating the bias correction status of these datasets in the original manuscript. (1) All datasets used in this study for driving BCC methods have undergone bias correction according to the standards of their respective data providers or have been subject to rigorous quality control procedures to ensure their accuracy and reliability. Therefore, we did not apply any further bias correction to these driving datasets ourselves; (2) For the datasets produced using both existing BCC methods and our proposed IWE-Res method in this study, no additional bias correction against observational data was performed. We added a discussion on the potential for further bias correction of these produced datasets by this study. Our specific revisions are as follows:

(1) We added descriptions of the bias correction and quality control for datasets driving BCC methods used in this study:

The GPCC precipitation dataset is derived from a large number of ground-based rain gauge observations collected globally and is produced through rigorous quality control and spatial interpolation procedures (Song et al., 2022; Wei et al., 2019; Abdelwares et al., 2020). The quality control process includes verification of station metadata (such as location and elevation), checks for temporal consistency, and the removal of extreme or erroneous values. As the GPCC dataset is interpolated from rain gauge observations and has undergone strict quality assurance, it is widely used in previous studies as a benchmark for bias correction of precipitation estimates from climate models.

The GPM IMERG Final Run precipitation product is a high-resolution, multi-source satellite-based precipitation dataset. It incorporates ground-based rain gauge observations and applies bias correction at the monthly scale (Wang et al., 2017; Cui et al., 2020; Huang et al., 2019).

The MSWEP dataset is produced by optimally merging precipitation data from satellite observations, ground stations, and reanalysis products (Beck et al., 2019; Beck et al., 2017). This dataset incorporates bias correction at the daily scale using more than 77,000 gauge observations worldwide (Beck et al., 2019; Beck et al., 2017).

The PERSIANN-CDR precipitation product is bias-corrected using the Global Precipitation Climatology Project (GPCP) 2.5° monthly product, which includes gauge data from the Global

Precipitation Climatology Centre (GPCC) (Chen et al., 2020; Kaprom et al., 2025; Sadeghi et al., 2019).

In contrast to precipitation, global ET datasets generally lack standardized and comprehensive bias correction procedures. Most bias correction approaches for ET are indirect, focusing on correcting the climate forcing inputs used to drive evapotranspiration models. This is primarily due to the limited availability of long-term, high-density in situ ET measurements globally—for example, the sparse distribution and limited representativeness of eddy covariance flux towers. The three ET products used in this study (GLDAS, GLEAM, and TerraClimate) improve data quality primarily through such indirect bias correction methods.

For TWSC, the GRACE TWSC datasets used in this study are provided by the University of Texas Center for Space Research (CSR), the German Research Centre for Geosciences (GFZ), and NASA's Jet Propulsion Laboratory (JPL), including multiple bias correction procedures to improve data quality (Landerer et al., 2012; Shamsudduha et al., 2017). These include filtering to suppress correlated noise and striping artifacts (Swenson et al., 2006), replacement of poorly resolved spherical harmonic coefficients (e.g., degree-2 term C20) with satellite laser ranging data (Loomis et al., 2020), and correction for glacial isostatic adjustment (GIA) (Peltier et al., 2012; Mu et al., 2017). Numerous studies have demonstrated the sensitivity and reliability of GRACE satellite data for monitoring TWSC (Swenson and Wahr, 2006; Resende et al., 2019; Majid et al., 2019; Reager et al., 2014).

The following sentences were added to the revised manuscript:

"Given the biases in the outputs of global P and ET models, observationally constrained datasets that have undergone bias correction or rigorous quality control are generally considered more accurate and reliable (Ehret et al., 2012). Accordingly, priority was given to datasets that incorporate extensive ground-based observations and provide bias-corrected or quality-controlled products. We selected four P datasets—GPCC, GPM IMERG, MSWEP, and PERSIANN-CDR; three ET datasets—GLDAS, GLEAM, and TerraClimate; and three TWSC datasets derived from GRACE satellite observations—GRACE CSR, GRACE GFZ, and GRACE JPL. All datasets were either bias-corrected according to the standards of their respective data providers or subjected to systematic quality control. Observed Q data were obtained from the GRDC platform. By combining these datasets, a total of 36 distinct data combinations were generated for each basin (Equation 3).

$$C_{jkl} = \begin{bmatrix} P_j & ET_k & TWSC_l & Q \end{bmatrix} \tag{2}$$

where $j$, $k$, and $l$ represent the indices of the datasets corresponding to each budget component. Table 1 provides basic information on the datasets used in this study, along with their corresponding indices. Equation 3 represents a matrix composed of the elements defined in Equation 2.

$$C = \begin{bmatrix} C_{111} & C_{112} & C_{113} & C_{121} & C_{122} & C_{123} & C_{131} & C_{132} & C_{133} \\ C_{211} & C_{212} & C_{213} & C_{221} & C_{222} & C_{223} & C_{231} & C_{232} & C_{233} \\ C_{311} & C_{312} & C_{313} & C_{321} & C_{322} & C_{323} & C_{331} & C_{332} & C_{333} \\ C_{411} & C_{412} & C_{413} & C_{421} & C_{422} & C_{423} & C_{431} & C_{432} & C_{433} \end{bmatrix} \tag{3}$$

The Global Precipitation Climatology Centre (GPCC) dataset, provided by the German Weather Service (DWD), is derived from a dense global network of rain gauge observations, and incorporates strict quality control procedures such as station data validation, temporal consistency checks, and outlier removal (Becker et al., 2013; Schneider et al., 2008). The dataset is available at 0.25° spatial resolution and daily to monthly temporal scales. The Global Precipitation Measurement Integrated Multi-Satellite Retrievals (GPM IMERG) Final Run product, developed

by NASA and JAXA, integrates multiple satellite-based precipitation estimates and applies monthly bias correction using ground-based gauge data (Wang et al., 2017; Cui et al., 2020; Huang et al., 2019). The Multi-Source Weighted-Ensemble Precipitation (MSWEP) dataset combines satellite, gauge, and reanalysis data using an ensemble-weighted approach, incorporating over 77,000 ground stations for daily-scale bias correction (Beck et al., 2019; Beck et al., 2017). The PERSIANN-CDR dataset, based on satellite remote sensing and artificial neural network technology, spans 60°S to 60°N with 0.25° daily resolution, and is bias-corrected using the GPCP monthly product, which includes extensive rain gauge observations (Chen et al., 2020; Kaprom et al., 2025; Sadeghi et al., 2019).

For ET, the Global Land Data Assimilation System (GLDAS), developed by NASA and NOAA, uses land surface modeling and data assimilation to produce physically consistent estimates of land surface fluxes. The GLEAM dataset, developed by the Miralles team at the University of Bristol, estimates actual ET using satellite-derived net radiation and air temperature via the Priestley-Taylor model, and applies a stress factor derived from vegetation optical depth (VOD) and soil moisture to adjust potential evaporation. TerraClimate dataset provides global monthly actual ET estimates based on the Penman Montieth approach (Abatzoglou et al., 2018). Notably, bias correction in global ET products is generally less systematic than for P products, mainly due to the limited availability and spatial coverage of in situ flux tower observations. As a result, bias adjustments in ET datasets are typically indirect, relying on corrections applied to the climate forcing variables rather than to ET itself.

The launch of the GRACE and GRACE Follow-On (GRACE-FO) satellite missions has provided new opportunities for more accurate observations of large-scale TWSC. GRACE operated from 2002 to 2017, followed by GRACE-FO starting in 2018 (Boergens et al., 2024). These missions infer terrestrial total TWSC by tracking temporal variations in Earth's gravity field, which are primarily attributed to changes in terrestrial water mass. The GRACE TWSC datasets used in this study are provided by the University of Texas Center for Space Research (CSR), the German Research Centre for Geosciences (GFZ), and NASA's Jet Propulsion Laboratory (JPL), all of which include multiple bias correction procedures to improve data quality (Landerer et al., 2012; Shamsudduha et al., 2017). These bias correction procedures include filtration to suppress correlated noise and striping artifacts (Swenson et al., 2006), replacement of poorly resolved spherical harmonic coefficients (e.g., degree-2 term C20) with satellite laser ranging data (Loomis et al., 2020), and correction for glacial isostatic adjustment (GIA) (Peltier et al., 2012; Mu et al., 2017). Numerous studies have demonstrated the sensitivity and reliability of GRACE satellite data for monitoring TWSC (Swenson and Wahr, 2006; Resende et al., 2019; Majid et al., 2019; Reager et al., 2014).

The Global Runoff Data Centre (GRDC) provides the most comprehensive open-access river discharge data available worldwide, collected from national hydrological agencies. This dataset includes river streamflow measurements from over 10,000 stations across 159 countries (Su et al., 2024). To minimize the impact of missing data on the reliability of the results, hydrological stations were selected based on the criterion that missing values accounted for less than 10% of the total dataset. Linear interpolation was then applied to fill any remaining data gaps.".

(2) We discussed the potential for further bias correction of the budget corrected datasets generated using the proposed IWE-Res method, based on observational data, although this lies beyond the scope of the present study. The following sentence has been added to the revised

manuscript:

"It is worth noting that the datasets generated by both the existing BCC methods and the IWE-Res method proposed in this study were not further bias-corrected against independent observations. For basin-specific applications requiring higher reliability, we recommend additional bias correction.".

- The resolution of the spatial problem is crucial. It is well known that changes in resolution make employed variables (such as DEMs, as in Aziz et al., 2022) very different at the same location. How to tackle this point? Which resolution 'advice' for the reader? Any criterion? Otherwise, the errors will propagate in a sort of uncontrolled cascade.

**Response:** Thank you for this important comment. We fully agree that the spatial resolution differences among datasets for budget components can introduce uncertainty, especially when rescaling is required. We apologize for not clearly stating the spatial resolution used in our study. In this work, water budget closure was performed at the basin scale and at the monthly temporal resolution. Specifically, all gridded datasets of budget components were spatially and temporally aggregated to monthly and basin scales, respectively. Therefore, we did not resample all datasets to a common grid resolution; instead, we upscaled each variable directly to the basin scale. Previous studies on water budget closure have also focused primarily on the basin scale (e.g., Lehmann et al., 2022; Abolafia-Rosenzweig et al., 2023; Luo et al., 2023; Wang et al., 2014; Tan et al., 2022; Sahoo et al., 2011), mainly for the following reasons:

(1) Water budget closure correction at the grid scale is extremely complex, as it requires accurate quantification of all inflow and outflow terms. These may include precipitation, evapotranspiration, changes in water storage, lateral inflows and outflows, leakage losses, and human withdrawals or returns. Many of these components are difficult to observe or estimate directly, particularly lateral fluxes and leakage. Neglecting these terms introduces substantial uncertainty into grid-scale analyses; (2) As you correctly noted, merging datasets with differing spatial resolutions onto a common grid introduces additional sources of uncertainty, which can reduce the accuracy of water budget closure correction; (3) The GRACE satellite-derived TWSC data have a relatively coarse spatial resolution, which limits their applicability at the grid scale. Since TWSC is a critical component of the monthly water budget closure correction, it cannot be omitted. By aggregating GRACE data to the basin scale, random errors tend to cancel out, thereby improving accuracy; (4) Despite recent advances in remote sensing and the expansion of global observational networks, significant uncertainties remain in gridded hydrological datasets when applied at fine spatial scales, especially for evapotranspiration. Thus, conducting water budget closure correction at the basin scale remains a central focus of current research.

In the revised manuscript, on the one hand, we clarified the spatial resolution applied in our analysis as "The above datasets were upscaled to the basin and monthly scales using spatial and temporal averaging."; On the other hand, we added a detailed discussion of why the basin scale was selected for water budget closure correction in this study as "Notably, the choice of spatial resolution has a significant impact on the results (Aziz et al., 2022; Bormann et al., 2006; Senan et al., 2022). Following many previous studies (Lehmann et al., 2022; Abolafia-Rosenzweig et al., 2023; Luo et al., 2023; Wang et al., 2014; Tan et al., 2022; Sahoo et al., 2011), the BCC method in this study is also applied at the basin scale rather than the grid scale for the following reasons: 1) Achieving water budget closure at the grid scale is complex and challenging due to the difficulty of quantifying

all water flux and storage components flowing into and out of the grid, including P, ET, TWSC, lateral inflow and outflow, leakage losses, and human water withdrawals and returns. Several of these components, such as lateral flow and leakage, are poorly observed or highly uncertain, and their omission introduces substantial error; 2) The datasets of different variables have varying spatial resolutions, and resampling them to a common resolution introduces uncertainties, which in turn affect the accuracy of water budget closure correction; 3) The coarse spatial resolution of GRACE-derived TWSC data limits their applicability for water budget closure calculation at the grid scale. At monthly resolution, TWSC is a critical component and cannot be neglected. Averaging GRACE data to the basin scale helps reduce random errors by offsetting positive and negative biases, thereby increasing the reliability of water budget closure correction; 4) Despite advances in remote sensing and in situ observation networks, grid-scale uncertainties remain substantial for some budget components, such as ET. Basin-scale analysis therefore reduces uncertainty and improves the reliability of water budget closure correction results.".

- • From line 289 on, I think that the noise to afflict the Kalman Filter with should be pointed out. Which is the observation noise covariance (and its quantification for the employed variables)?

**Response:** Thank you for your careful review. We sincerely apologize for the insufficient explanation of the observation noise covariance and its quantification in the CKF method in the original manuscript. The CKF method used in this study is developed by Pan and Wood (2006) based on the Kalman Filter method. Given that the CKF method has been widely applied in previous studies for correcting budget component datasets, we included it as one of the benchmark methods to evaluate the performance of the proposed IWE-Res method.

In the CKF method, the reference values used to compute the error covariance ($\varepsilon$ in Equations 7–10) differ among the four budget components (P, ET, Q, and TWSC). For P, ET, and TWSC, due to the lack of spatially matched ground-truth observations at the grid scale, previous studies (Zhang et al., 2018; Abolafia-Rosenzweig et al., 2021) have typically used the ensemble mean across all datasets considered in their studies as the reference value. For Q, both previous studies and the present work use observed Q in the CKF method. Previous studies have reported gauge-based uncertainty as a percent error for some of the basins, ranging from 2.3% – 28.8% (Clarke, 1999; Mueller, 2003; Shiklomanov et al., 2006; Abolafia-Rosenzweig et al., 2021).

To ensure consistency with previous studies, we adopted the same assumptions in our application of the CKF method. However, as this method relies on approximated reference values, it may introduce inaccuracies in the error estimation for certain budget components. This, in turn, can propagate uncertainty into the corrected datasets. Hence, evaluating the uncertainty ranges of existing correction methods and developing new methods to reduce those uncertainties is essential. This concern forms a core motivation of our study: (1) to quantify the uncertainties introduced by existing BCC methods (CKF, MCL, MSD, and PR) at the monthly scale across 84 global basins spanning diverse climate zones; and (2) to propose a novel method, IWE-Res, for identifying the optimal balance in ΔRes redistribution, minimizing the combined error from both introduced budget component errors and the remaining ΔRes error.

In the revised manuscript, we carefully revised the description of the CKF method to clarify the relationships among the relevant equations and to more clearly explain the quantification of the error covariance for each budget component. Additionally, we incorporated more references to

previous studies that have applied the CKF method. The specific revisions made to the manuscript are as follows:

"The CKF method is developed based on the Kalman filter method (Pan and wood, 2006). For a given set of estimated budget components $X = [P\ ET\ Q\ TWSC]^T$ and their estimated errors $\Delta Res = GX \neq 0$ (where G is a constant vector, $G = [1\ -1\ -1\ -1]$), the goal is to find a new set of estimates $F = [P'\ ET'\ Q'\ TWSC']^T$ such that $GX' = 0$, achieving water budget closure (Pan et al., 2012). In simple terms, the CKF method redistributes the $\Delta Res$ among the budget components based on the error covariance of $X$, defined as $\Delta\varepsilon_{XX}$ (Equation 7), to obtain a closured dataset.

$$\Delta\varepsilon_{XX} = \overline{(X - X_0)(X - X_0)^T} \tag{7}$$

where $X_0$ refers to the reference values of the estimated budget components, and the bar over an expression denotes expectation. For P, ET and TWSC, the reference values $X_0$ were calculated by averaging all considered datasets, following previous studies (Zhang et al., 2018; Abolafia-Rosenzweig et al., 2021). For Q, we adopted observed Q. Due to the difficulty in quantifying the uncertainty in observed Q, previous studies have reported gauge-based uncertainty as a percent error for some of the basins, ranging from 2.3%–28.8% (Clarke, 1999; Mueller, 2003; Shiklomanov et al., 2006; Abolafia-Rosenzweig et al., 2021). We followed a similar approach to estimate the uncertainty associated with Q in this study.

The error covariance matrix $\Delta\varepsilon_{XX}$ is of dimension 4×4 and represents the covariances among errors in the four budget components:

$$\Delta\varepsilon_{XX} = \begin{bmatrix} \Delta\varepsilon_{P-P} & \Delta\varepsilon_{P-ET} & \Delta\varepsilon_{P-Q} & \Delta\varepsilon_{P-TWSC} \\ \Delta\varepsilon_{ET-P} & \Delta\varepsilon_{ET-ET} & \Delta\varepsilon_{ET-Q} & \Delta\varepsilon_{ET-TWSC} \\ \Delta\varepsilon_{Q-P} & \Delta\varepsilon_{Q-ET} & \Delta\varepsilon_{Q-Q} & \Delta\varepsilon_{Q-TWSC} \\ \Delta\varepsilon_{TWSC-P} & \Delta\varepsilon_{TWSC-ET} & \Delta\varepsilon_{TWSC-Q} & \Delta\varepsilon_{TWSC-TWSC} \end{bmatrix} \tag{8}$$

Following Pan et al. (2012), the off-diagonal elements representing cross-variable error covariances were assumed to be zero, under the assumption that errors among different budget components are uncorrelated. Accordingly, the matrix F can be computed as shown in Equation 9.

$$F = X + K(0 - GX) \tag{9}$$

where $K = \Delta\varepsilon_{XX}C^T(C\Delta\varepsilon_{XX}C^T)^{-1}$ is the Kalman gain. Setting $GX = \Delta Res$, and Equation 9 can be rewritten as Equation 10.

$$F = X - \Delta\varepsilon_{XX}G^T(G\Delta\varepsilon_{XX}G^T)^{-1}\Delta Res \tag{10}$$

where error covariance $\varepsilon_{XX}$ is calculated entry by entry according to Equation 8.".

- Is there any exit criterion (such as tolerances) to exit from equations 9 and 10?

**Response:** Thank you for your careful review. The existing CKF method uses the total number of months within the study period as the only termination criterion for iteration—that is, it runs for a fixed number of iterations corresponding to the study duration. No additional exit conditions are set for Equations 9 and 10. This design is rooted in the core principle of the CKF method: it first estimates the error covariances of the water budget components and then proportionally redistributes the water imbalance (ΔRes) back to the raw data based on these estimated errors. However, if the error covariances are inaccurately estimated, the redistribution of ΔRes may be suboptimal. This misallocation can then propagate: an excessive correction to one variable reduces the remaining ΔRes available for others, potentially resulting in further inaccuracies. Notably, this iteration process continues until the predetermined number of steps (the total number of months) is reached,

regardless of whether such misallocations occur.

We highlighted this limitation in the modified manuscript by stating: "Beyond introducing negative values, such imbalanced redistribution compromises the integrity of the remaining components. Overcorrecting one variable necessarily reduces the share of ∆ Res available for others, potentially degrading their accuracy.".

To address these limitations of existing methods, we proposed the IWE-Res method. Unlike existing methods, IWE-Res introduces exit conditions based on the physical plausibility of the corrected variables. Specifically, we terminate the iteration if any of the following occurs: P, ET, or Q becomes negative, or if the sign of TWSC reverses (from positive to negative or vice versa). The following sentences were added to the revised manuscript as: "During the iterative correction process, if any of the water budget components (P, ET, and Q) becomes negative, the redistribution of water imbalance error to that component is immediately suspended. In subsequent iterations, redistribution is recalculated to ensure that only components with physically meaningful positive values receive the imbalance correction. For example, if ET becomes negative in a given iteration, the imbalance is subsequently redistributed to P, Q, and TWSC only, in accordance with Equation 33. For TWSC, if a sign reversal occurs during iteration (i.e., from positive to negative or vice versa), the redistribution of the water imbalance error to TWSC is suspended in the following iteration.".

Suggested References:
- Levison et al., 2016. Long-term trends in groundwater recharge and discharge in a fractured bedrock aquifer – past and future conditions. Canadian Water Resources Journal / Revue canadienne des ressources hydriques, Volume 41, 2016 - Issue 4: Special Issue: Groundwater – Surface Water Interactions in Canada. https://doi.org/10.1080/07011784.2015.1037795
- Schiavo, 2023. The role of different sources of uncertainty on the stochastic quantification of subsurface discharges in heterogeneous aquifers. J. Hydrol. 617 (4), 128930. DOI: 10.1016/j.jhydrol.2022.128930

Further reading references:
- Ehret, U., Zehe, E., Wulfmeyer, V., Warrach-Sagi, K., and Liebert, J.: HESS Opinions "Should we apply bias correction to global and regional climate model data?", Hydrol. Earth Syst. Sci., 16, 3391–3404, https://doi.org/10.5194/hess-16-3391-2012, 2012.
- Aziz, K.M.A., Rashwan, K.S. Comparison of different resolutions of six free online DEMs with GPS elevation data on a new 6th of October City, Egypt. *Arab J Geosci* **15**, 1585 (2022). https://doi.org/10.1007/s12517-022-10845-5

**Response:** Thanks very much for the useful references. We carefully reviewed the suggested references and incorporated them into the revised manuscript.

**References:**

Abatzoglou, J. T., Dobrowski, S. Z., Parks, S. A., & Hegewisch, K. C. (2018). TerraClimate, a high-resolution global dataset of monthly climate and climatic water balance from 1958–2015. Scientific Data, 5(1), 1-12. https://doi.org/10.1038/sdata.2017.191

Abdelwares, M., Lelieveld, J., Zittis, G., Haggag, M., & Wagdy, A. (2020). A comparison of gridded datasets of precipitation and temperature over the Eastern Nile Basin region. Euro-Mediterranean Journal for Environmental Integration, 5, 1-16. https://doi.org/10.1007/s41207-

019-0140-y

Abolafia-Rosenzweig, R., Pan, M., Zeng, J. L., & Livneh, B. (2021). Remotely sensed ensembles of the terrestrial water budget over major global river basins: An assessment of three closure techniques. Remote Sensing of Environment, 252, 112191. https://doi.org/10.1016/j.rse.2020.112191

Aziz, K. M. A., & Rashwan, K. S. (2022). Comparison of different resolutions of six free online DEMs with GPS elevation data on a new 6th of October City, Egypt. Arabian Journal of Geosciences, 15(20), 1585. https://doi.org/10.1007/s12517-022-10845-5

Beck, H. E., Vergopolan, N., Pan, M., Levizzani, V., Van Dijk, A. I., Weedon, G. P., . . . Wood, E. F. (2017). Global-scale evaluation of 22 precipitation datasets using gauge observations and hydrological modeling. Hydrology and Earth System Sciences, 21(12), 6201-6217. https://doi.org/10.5194/hess-21-6201-2017

Beck, H. E., Wood, E. F., Pan, M., Fisher, C. K., Miralles, D. G., Van Dijk, A. I., . . . Adler, R. F. (2019). MSWEP V2 global 3-hourly 0.1 precipitation: methodology and quantitative assessment. Bulletin of the American Meteorological society, 100(3), 473-500. https://doi.org/10.1175/BAMS-D-17-0138.1

Becker, A., Finger, P., Meyer-Christoffer, A., Rudolf, B., Schamm, K., Schneider, U., & Ziese, M. (2013). A description of the global land-surface precipitation data products of the Global Precipitation Climatology Centre with sample applications including centennial (trend) analysis from 1901–present. Earth System Science Data, 5(1), 71-99. https://doi.org/10.5194/essd-5-71-2013

Boergens, E., Güntner, A., Sips, M., Schwatke, C., & Dobslaw, H. (2024). Interannual variations of terrestrial water storage in the East African Rift region. Hydrology and Earth System Sciences, 28(20), 4733-4754. https://doi.org/10.5194/hess-28-4733-2024

Bormann, H. (2006). Impact of spatial data resolution on simulated catchment water balances and model performance of the multi-scale TOPLATS model. Hydrology and Earth System Sciences, 10(2), 165-179. https://doi.org/10.5194/hess-10-165-2006

Chen, S., Liu, B., Tan, X., & Wu, Y. (2020). Inter-comparison of spatiotemporal features of precipitation extremes within six daily precipitation products. Climate Dynamics, 54, 1057-1076. https://doi.org/10.1007/s00382-019-05045-z

Clarke, R. T. (1999). Uncertainty in the estimation of mean annual flood due to rating-curve indefinition. Journal of Hydrology, 222(1-4), 185-190. https://doi.org/10.1016/S0022-1694(99)00097-9

Cui, W., Dong, X., Xi, B., Feng, Z. H. E., & Fan, J. (2020). Can the GPM IMERG final product accurately represent MCSs' precipitation characteristics over the central and eastern United States?. Journal of Hydrometeorology, 21(1), 39-57. https://doi.org/10.1175/JHM-D-19-0123.1

Ehret, U., Zehe, E., Wulfmeyer, V., & Liebert, J. (2012). Should we apply bias correction to global and regional climate model data? HESS, 16, 3391–3404. https://doi.org/10.5194/hess-16-3391-2012, 2012.

Huang, C., Hu, J., Chen, S., Zhang, A., Liang, Z., Tong, X., ... & Zhang, Z. (2019). How well can IMERG products capture typhoon extreme precipitation events over southern China?. Remote Sensing, 11(1), 70. https://doi.org/10.3390/rs11010070

Kaprom, C., Williams, J. A., Mehrotra, R., Ophaphaibun, C., & Sriwongsitanon, N. (2025). A

comprehensive evaluation of the accuracy of satellite-based precipitation estimates over Thailand. Journal of Hydrology: Regional Studies, 59, 102380. https://doi.org/10.1016/j.ejrh.2025.102380

Landerer, F. W., & Swenson, S. C. (2012). Accuracy of scaled GRACE terrestrial water storage estimates. Water resources research, 48(4). https://doi.org/10.1029/2011WR011453

Lehmann, F., Vishwakarma, B. D., & Bamber, J. (2022). How well are we able to close the water budget at the global scale?. Hydrology and Earth System Sciences, 26(1), 35-54. https://doi.org/10.5194/hess-26-35-2022

Levison, J., Larocque, M., Ouellet, M. A., Ferland, O., & Poirier, C. (2016). Long-term trends in groundwater recharge and discharge in a fractured bedrock aquifer–past and future conditions. Canadian Water Resources Journal/Revue canadienne des ressources hydriques, 41(4), 500-514. https://doi.org/10.1080/07011784.2015.1037795

Loomis, B. D., Rachlin, K. E., Wiese, D. N., Landerer, F. W., & Luthcke, S. B. (2020). Replacing GRACE/GRACE‐FO with satellite laser ranging: Impacts on Antarctic Ice Sheet mass change. Geophysical Research Letters, 47(3), e2019GL085488. https://doi.org/10.1029/2019GL085488

Luo, Z., Li, H., Zhang, S., Wang, L., Wang, S., & Wang, L. (2023). A novel two‐step method for enforcing water budget closure and an intercomparison of budget closure correction methods based on satellite hydrological products. Water Resources Research, 59(3), e2022WR032176. https://doi.org/10.1029/2022WR032176

Majid, R., & Ardalan, E. S. (2019). Performance of the Gravity Recovery and Climate Experiment (GRACE) method in monitoring groundwater-level changes in local-scale study regions within Iran. Hydrogeology Journal, 27(7), 2497-2509. https://doi.org/10.1007/s10040-019-02007-x

Mehrnegar, N., Schumacher, M., Jagdhuber, T., & Forootan, E. (2023). Making the best use of GRACE, GRACE‐FO and SMAP data through a constrained Bayesian data‐model integration. Water Resources Research, 59(9), e2023WR034544. https://doi.org/10.1029/2023WR034544

Mu, D., Yan, H., Feng, W., & Peng, P. (2017). GRACE leakage error correction with regularization technique: Case studies in Greenland and Antarctica. Geophysical Journal International, 208(3), 1775-1786. https://doi.org/10.1093/gji/ggw494

Mueller, D. S. (2003, March). Field evaluation of boat-mounted acoustic Doppler instruments used to measure streamflow. In Proceedings of the IEEE/OES Seventh Working Conference on Current Measurement Technology, 2003. (pp. 30-34). IEEE. https://doi.org/10.1109/CCM.2003.1194278

Pan, M., Sahoo, A. K., Troy, T. J., Vinukollu, R. K., Sheffield, J., & Wood, E. F. (2012). Multisource estimation of long-term terrestrial water budget for major global river basins. Journal of Climate, 25(9), 3191-3206. https://doi.org/10.1175/JCLI-D-11-00300.1

Pellet, V., Aires, F., Papa, F., Munier, S., & Decharme, B. (2020). Long-term total water storage change from a Satellite Water Cycle reconstruction over large southern Asian basins. Hydrology and Earth System Sciences, 24(6), 3033-3055. https://doi.org/10.5194/hess-24-3033-2020

Peltier, W. R., Drummond, R., & Roy, K. (2012). Comment on "Ocean mass from GRACE and glacial isostatic adjustment" by DP Chambers et al. Journal of Geophysical Research: Solid Earth, 117(B11). https://doi.org/10.1029/2011JB008967

Reager, J. T., Thomas, B. F., & Famiglietti, J. S. (2014). River basin flood potential inferred using GRACE gravity observations at several months lead time. Nature Geoscience, 7(8), 588-592. https://doi.org/10.1038/ngeo2203

Resende, T. C., Longuevergne, L., Gurdak, J., Leblanc, M., Favreau, G., Ansems, N., ... & Aureli, A. (2019). Assessment of the impacts of climate variability on total water storage across Africa: implications for groundwater resources management. Hydrogeology Journal, 27(1), 493-512. https://doi.org/10.1007/s10040-018-1864-5

Sadeghi, M., Akbari Asanjan, A., Faridzad, M., Afzali Gorooh, V., Nguyen, P., Hsu, K., ... & Braithwaite, D. (2019). Evaluation of PERSIANN-CDR constructed using GPCP V2. 2 and V2. 3 and a comparison with TRMM 3B42 V7 and CPC unified gauge-based analysis in global scale. Remote Sensing, 11(23), 2755. https://doi.org/10.3390/rs11232755

Sahoo, A. K., Pan, M., Troy, T. J., Vinukollu, R. K., Sheffield, J., & Wood, E. F. (2011). Reconciling the global terrestrial water budget using satellite remote sensing. Remote Sensing of Environment, 115(8), 1850-1865. https://doi.org/10.1016/j.rse.2011.03.009

Schiavo, M. (2023). The role of different sources of uncertainty on the stochastic quantification of subsurface discharges in heterogeneous aquifers. Journal of Hydrology, 617, 128930. https://doi.org/10.1016/j.jhydrol.2022.128930

Schneider, U., Fuchs, T., Meyer-Christoffer, A., & Rudolf, B. (2008). Global precipitation analysis products of the GPCC. Global Precipitation Climatology Centre (GPCC), DWD, Internet Publikation, 112, 3819-3837.

Senan, S., Thomas, J., Vema, V. K., Jainet, P. J., Nizar, S., Sivan, S., & Sudheer, K. P. (2022). A study of the influence of rainfall datasets' spatial resolution on stream simulation in Chaliyar River Basin, India. Journal of Water and Climate Change, 13(12), 4234-4254. https://doi.org/10.2166/wcc.2022.273

Shamsudduha, M., Taylor, R. G., Jones, D., Longuevergne, L., Owor, M., & Tindimugaya, C. (2017). Recent changes in terrestrial water storage in the Upper Nile Basin: an evaluation of commonly used gridded GRACE products. Hydrology and Earth system sciences, 21(9), 4533-4549. https://doi.org/10.5194/hess-21-4533-2017

Shiklomanov, A. I., Yakovleva, T. I., Lammers, R. B., Karasev, I. P., Vörösmarty, C. J., & Linder, E. (2006). Cold region river discharge uncertainty—Estimates from large Russian rivers. Journal of Hydrology, 326(1–4), 231–256. https://doi.org/10.1016/j.jhydrol.2005.10.037

Song, L., Xu, C., Long, Y., Lei, X., Suo, N., & Cao, L. (2022). Performance of seven gridded precipitation products over arid central Asia and subregions. Remote Sensing, 14(23), 6039. https://doi.org/10.3390/rs14236039

Su, Y., & Zhang, S. (2024). Optimizing Parameters in the Common Land Model by Using Gravity Recovery and Climate Experiment Satellite Observations. Land, 13(4), 508. https://doi.org/10.3390/land13040508

Swenson, S., & Wahr, J. (2006). Post‑processing removal of correlated errors in GRACE data. Geophysical research letters, 33(8). https://doi.org/10.1029/2005GL025285

Tan, X., Liu, B., Tan, X., & Chen, X. (2022). Long‑term water imbalances of watersheds resulting from biases in hydroclimatic data sets for water budget analyses. Water Resources Research, 58(3), e2021WR031209. https://doi.org/10.1029/2021WR031209

Wang, L., Wang, J., Li, M., Wang, L., Li, X., & Zhu, L. (2022). Response of terrestrial water storage and its change to climate change in the endorheic Tibetan Plateau. Journal of Hydrology, 612,

128231. https://doi.org/10.1016/j.jhydrol.2022.128231

Wang, S., McKenney, D. W., Shang, J., & Li, J. (2014). A national‑scale assessment of long‑term water budget closures for Canada's watersheds. Journal of Geophysical Research: Atmospheres, 119(14), 8712-8725. https://doi.org/10.1002/2014JD021951

Wang, Z., Zhong, R., Lai, C., & Chen, J. (2017). Evaluation of the GPM IMERG satellite-based precipitation products and the hydrological utility. Atmospheric Research, 196, 151-163. https://doi.org/10.1016/j.atmosres.2017.06.020

Wei, L., Jiang, S., Ren, L., Yuan, F., & Zhang, L. (2019). Performance of two long-term satellite-based and GPCC 8.0 precipitation products for drought monitoring over the Yellow River Basin in China. Sustainability, 11(18), 4969. https://doi.org/10.3390/su11184969

Zhang, Y., Pan, M., Sheffield, J., Siemann, A. L., Fisher, C. K., Liang, M., . . . Houser, P. R. (2018). A Climate Data Record (CDR) for the global terrestrial water budget: 1984–2010. Hydrology and Earth System Sciences, 22(1), 241-263. https://doi.org/10.5194/hess-22-241-2018

---

## Author Comment (AC4)

Response to Review
Reviewer #1 (RC2):

Thanks to the authors team for a careful response to the two comments. From the text revisions that you show I can see that all substantive points have been adequately addressed and that the manuscript will now be an important contribution to the literature. Congratulations.

**Response**: Thank you very much for your positive and constructive review, which helped us to improve the manuscript greatly. Thank you again.